

# The impact of structural error on parameter constraint in a climate model

Doug McNeall[1], Jonny Williams[2], Ben Booth[1], Richard Betts[1], Peter Challenor[3], Andy Wiltshire[1], and David Sexton[1]

[1]Met Office Hadley Centre, FitzRoy Road, Exeter, EX1 3PB UK
[2]NIWA, 301 Evans Bay Parade, Hataitai, Wellington 6021, New Zealand
[3]University of Exeter, North Park Road, Exeter EX4 4QE UK

*Correspondence to:* Doug McNeall (doug.mcneall@metoffice.gov.uk)

**Abstract.** We use observations of forest fraction to constrain carbon cycle and land surface input parameters of the reduced resolution global climate model, FAMOUS. Using a history matching approach along with a computationally cheap statistical proxy (emulator) of the climate model, we compare an ensemble of simulations of forest fraction with observations, and rule out parameter settings where the forests are poorly simulated.

Regions of parameter space where FAMOUS best simulates the Amazon forest fraction are incompatible with the regions where FAMOUS best simulates other forests, indicating a structural error in the model. Using observations of the Amazon forest to constrain input parameters leads to very different conclusions about the acceptable values of input parameters than using the other forests.

      We characterise the structural model discrepancy, and explore the consequences of ignoring it in a history matching exercise.
We use sensitivity analysis to find the parameters which have most impact on simulator error. We use the emulator to simulate the forest fraction at the best set of parameters implied by matching the model to the Amazon, and to other major forests in turn. We can find parameters that lead to a realistic forest fraction in the Amazon, but using the Amazon alone to tune the simulator would result in a significant overestimate of forest fraction in the other forests. Conversely, using the other forests to calibrate the model leads to a larger underestimate of the Amazon forest fraction.

Finally, we perform a history matching exercise using credible estimates for simulator discrepancy and observational uncertainty terms. We are unable to constrain the parameters individually, but just under half of joint parameter space is ruled out as being incompatible with forest observations. We discuss the possible sources of the discrepancy in the simulated Amazon, including missing processes in the land surface component, and a bias in the climatology of the Amazon.

## 1   Introduction

A common practice in Earth system modelling is the parameterisation of processes which are too computationally expensive to represent explicitly. These parameterisations have associated numerical coefficients, quantitatively representing some process. The coefficients may directly represent a measurable physical quantity, or they may be a more abstract representation necessary due to the simplification of the modelled process. There is often uncertainty about the value of the parameter coefficients that





should be used to best represent the system being simulated. It may not be desirable or practical to choose a single value of the coefficients over all others, and uncertainty in the best choice of parameters can be represented by using a range of values for each of the coefficients in an ensemble of simulator runs.

Choosing appropriate values of these coefficients is a major research effort that encompasses domain specific, statistical and computational literature. The coefficients are tuneable by comparison of the behaviour of the simulator with observations of the real system, although there may also be direct measurements of the value of the coefficient or other information from theory. There is a long history of using observations to constrain parameterisation coefficients within General Circulation Models (GCMs), particularly within atmospheric components. Where this is done as an inverse problem in formal probabilistic setting, then it may also provide probability distributions for the parameters of the model, and is known as *calibration*. The process of

choosing a single best parameter set is often called *tuning*. *History matching* provides a formal way of ruling out parameter settings that are inconsistent with observed data.

The motivation for calibration of a simulator is twofold. First, a simulator which matches the underlying dynamics of a system better should produce more accurate predictions. Second, given an accurate simulator, a more tightly constrained parameter set should provide a narrower range of uncertainty in future predictions.

## 1.1  Simulator discrepancy

Simulator discrepancy is the systematic difference between a climate model, or simulator, and the system that is represented by that model. It can also be known as model (or simulator) bias, model error, or structural error. A useful definition from Williamson et al. (2014) is that *"A climate model bias [simulator discrepancy] represents a structural error if that bias cannot be removed by changing the parameters without introducing more serious biases to the model"*. One of the main aims of the

model development process is to efficiently identify important simulator discrepancies and correct them, or allow them to be taken into account in analyses; for example, during prediction using the simulator (e.g. Sexton et al. (2011)).

Simulator discrepancy is a major challenge during calibration. In many cases, there is an indeterminacy between parameter error and simulator discrepancy; that is, should we choose a different set of parameters as representing the "best" or should we add a simulator discrepancy term? Sometimes, there is little or no information to distinguish between these two.

Simulator discrepancy might be known a priori - perhaps a computationally necessary simplification or parameterisation, has a predictable effect on simulator output. Alternatively, the discrepancy might be due to some missing and unknown process in the model. This sort of discrepancy might appear as a bias, and only become apparent when output from the simulator is compared with observations of the phenomena under study in the real system. In both cases, the modeller must have a strategy for dealing with the discrepancy when using the simulator to make judgements about the system.

Kennedy and O'Hagan (2001) introduced a Bayesian framework to the task of the calibration of computationally expensive simulators. They urge the specification of a priori estimates of simulator discrepancy, and offer methods to learn about that discrepancy by comparison of the simulator and observations. Failure to take model discrepancy into account in calibration can lead to overconfident and inaccurate estimates of the parameters, and consequently the predictions of the model (e.g.





Brynjarsdóttir and O'Hagan (2014), Higdon et al. (2008)). Further, even inadequate (as opposed to outright wrong) specification of a simulator discrepancy can lead to overconfidence and bias in parameters and predictions.

## 1.2 Calibration of Land surface components

Parametric uncertainty in the land surface and carbon cycle component of models is expected to represent a large fraction of
current uncertainty in future climate projections (Booth et al. (2012), Booth et al. (2013), Huntingford et al. (2009)). These components have been introduced into climate models more recently, and have not yet been subject to the depth of systematic evaluation as, for example, atmospheric components. There is much focus therefore, in identifying parameter sets that are consistent with observed climate metrics, or at least reducing future land carbon cycle uncertainty by identifying which parts of possible model parameter space are inconsistent with observed properties of the real climate system.

There is also a long history of statistical and data assimilation approaches used to constrain process model parameters. In the land surface model context these extend back to Sellers et al. (1996). Recent examples are community efforts to develop a systematic set of observations to benchmark land surface processes against metrics of real world processes, for example the International Land Model Benchmarking Project (Luo et al., 2012), and PALS (Abramowitz, 2012). Such benchmarks involve an extensive set of metrics, covering a broad cross-section of model processes. These benchmarks enable an assessment of
overall model skill and highlight particular areas where the model falls short. They provide a useful framework to assess improvements in model skill that arise from continual model development as well as prioritising resources towards model processes that are less well simulated. Using a large number of observed metrics for diverse aspects of the model processes also helps avoid model parameters being tuned to address a particular process, to the detriment of wider model performance. One of the limitations of the benchmarking approach is that there is only limited current understanding of what information
a given observed metric implies about the model formulation or parameters, or what this might imply about future projected changes.

## 1.3 Paper aims and outline

Our aim is to identify parameter sets for the land surface module of the climate simulator FAMOUS where the simulator output and the observations of forest fraction are consistent to an acceptable degree. An initial attempt using history matching suggests
that FAMOUS is unable to simulate the Amazon forest and other forests simultaneously at any set of parameters within the experiment design. We argue that this is due to a fundamental simulator discrepancy, which has implications for constraining the input parameters of FAMOUS. We use a number of techniques to characterise and find the drivers of this structural error, before performing a second history match with an appropriate discrepancy function.

In Sect. 2 we briefly describe the ensemble of a climate simulator, and describe the emulator and the history matching
technique that we use to explore simulator discrepancy in Sect. 2.5 and 2.6 respectively. We perform an initial history matching exercise in Sect. 3.1. We use the emulator to quantify the relationships between the simulated forest fraction and a set of model input parameters in a sensitivity analysis in Sect. 3.2. Next, we measure the performance of the model ensemble in simulating forest fraction in Sect. 3.3. We see how much input space would be ruled out as implausible in various scenarios of data





combination and uncertainty budget in Sect. 3.4 and we learn what each individual observation tells us about input space in Sect. 3.5. In Sect. 3.6, we use the emulator and an implausibility measure to find the "best" set of parameters for each forest, and project the consequences of using those parameters on the other forests. Finally, we perform a history matching exercise with a credible discrepancy function to constrain input parameters in Sect. 3.7. In Sect. 4, we discuss the consequences of our

findings for simulators of the Amazon rainforest. We offer conclusions in Sect. 5.

## 2  Data and Methods

### 2.1  The FAMOUS climate model

We use a pre-existing ensemble of the climate model FAMOUS throughout this study. The Fast Met Office UK Universities Simulator FAMOUS (Jones et al., 2005; Smith et al., 2008) is a reduced resolution climate simulator, based on, and tuned to

replicate, the climate model HadCM3 (Gordon et al., 2000; Pope et al., 2000). Computational efficiency is gained primarily through reduced resolution. Atmospheric grid boxes are four times the size of HadCM3, and ocean gridboxes are also larger. There are fewer levels in the atmosphere (11 compared to 19), and the ocean timestep is 12 hours compared to 1 hour for HadCM3. In the atmosphere, the timestep is 1 hour, doubled from HadCM3. The dynamic vegetation component is called TRIFFID and is described in detail in Cox (2001). FAMOUS runs approximately ten times faster than HadCM3, making it

ideal for running large ensembles, or long integrations, with modest supercomputing facilities.

Smith (2012) describe improvements to FAMOUS in sea ice, ozone, hydrological cycle conservation and upper tropospheric dynamics. Williams et al. (2013) describe the inclusion of the carbon cycle in the model via perturbed physics ensembles of terrestrial and ocean parameters, of which the terrestrial ensemble is studied in this paper. Most recently, Williams et al. (2014) give details of inclusion of a scheme to simulate the cycling of oxygen in the ocean and its coupling with the carbon cycle.

The explicit inclusion of vegetation in FAMOUS is documented in Williams et al. (2013), which introduces surface tiling in the newer MOSES2 scheme. Five different vegetation types are simulated: broadleaf and needleleaf trees, C3 and C4 grasses, and shrubs, each with a fractional coverage in a gridbox. Several surface types represent the absence of vegetation: bare soil, land ice, urbanised land use and inland water. Williams et al. (2013) describe the optimisation of carbon cycle parameters in the terrestrial and ocean domains, validated against observations and reanalysis products, and present climatologies using both

fixed and dynamic vegetation.

### 2.2  Known biases in the climate of FAMOUS

FAMOUS shows a northern-hemisphere-winter surface air temperature cold bias with respect to HadCM3 and also the over-estimation of the fractions of needleleaf trees in North America and C3 grassland in the northern part of Eurasia. The initial version of FAMOUS, used the MOSES1 surface exchange scheme, and did not explicitly describe the inclusion of any veg-

etation cover, instead using gridbox averages of surface quantities such as root depth, surface albedo and roughness length to describe momentum and water exchange between the surface and the atmosphere. Biases were already present in climate



**Table 1.** Land surface input parameters for FAMOUS

| Parameter | Description |
| --- | --- |
| F0 | Ratio of $CO_2$ concentrations inside and outside leaves at zero humidity deficit. |
| LAI_MIN | PFT must achieve this value of the leaf area index before it starts to contend with other PFTs for growing area. |
| NL0 | Top leaf nitrogen concentration. The amount of nitrogen per amount of carbon. |
| R_GROW | Growth respiration fraction. |
| TUPP | Control on variation of photosynthesis with temperature. |
| Q10 | Control on soil respiration with temperature. |
| V_CRIT_ALPHA | Control of photosynthesis with soil moisture. |

regimes (Gnanadesikan and Stouffer, 2006) relevant for the Amazon rainforest. Smith et al. (2008) noted: *"the Amazon region is not wet enough for a fully humid region to exist."*

## 2.3 The ensemble

We use an ensemble of 100 simulations of FAMOUS detailed in Williams et al. (2013), and build upon the results of that study. The ensemble was run in order to test the utility of including the carbon cycle in enhancing the FAMOUS model. The ensemble design perturbs 7 vegetation and land surface control parameters (see table 1) in a latin hypercube configuration (McKay et al., 1979). This kind of design efficiently spans parameter space, and has been shown to be better than others for constructing surface response type statistical models known as emulators (Urban and Fricker, 2010).

This design builds upon a previous ensemble run by Gregoire et al. (2010), and implicitly contains a further parameter, $\beta$, that indexes into that other ensemble. The $\beta$ parameter indexes the top 10 performing models with regards to the atmospheric climate. The Beta parameter is uncorrelated with any land surface parameters and the model output, so we exclude it from the ensemble design, essentially treating it as a nuisance parameter.

Ranges for the land surface parameters follow those used in the study by Booth et al. (2012), and as that paper makes clear were chosen for a number of reasons, not necessarily to represent plausible ranges of their uncertainty. However, we are confident that the parameter ranges are wide enough to span the space which might a priori be considered reasonable.

The ensemble simulates the preindustrial climate, with ensemble members spun up over a 200 year period to ensure that the vegetation is in equilibrium with the climate at 290 ppm of $CO_2$. The vegetation dynamics component of the simulator, TRIFFID is run for the equivalent of 10,000 years for each decade of climate to allow for the long adjustment time of dynamic vegetation. The climatology is constructed using the final 30 year period of the ensemble.



## 2.4 Simulator outputs and observations

We use forest fraction as the primary simulator output for study. Observations of forest fraction were adapted from Loveland et al. (2000), and are regionally aggregated versions of the data used in the previous study by Williams et al. (2013). We use broadleaf only for the tropical forest, and a mixture of broadleaf and needleleaf for the North American forest. A summary

of the forest fraction data in the ensemble can be found in figures 1 and 2. The former shows the spatial distribution of forest fraction in FAMOUS in maps of both the mean and standard deviation across the ensemble of 100 members. Parameter ranges are not explicitly chosen to represent uncertainty, and so the ensemble mean and standard deviation are not a meaningful representation of data uncertainty but provide a useful summary of the data. To summarise the forest fraction data, we find the mean forest fraction in each of the Amazon, Central African, South East Asian, North American and Global regions (see

supplementary matterial Fig. S1 for region details).

Figure 2 shows every input and summary output, plotted against each other. This shows the marginal relationships of the 1) inputs against the inputs (which as expected show no obvious relationship), 2) the strength of the marginal relationship between the inputs and outputs, and 3) the outputs against the outputs, which highlights where outputs vary together.

South East Asian and Central African forests vary together very strongly across the ensemble, whereas the Central African

and North American forests show a weaker relationship, with more scatter. This might be expected, given the different structure of the North American forests, compared with the tropical. The scatter plot also identifies NL0 (leaf Nitrogen) and V_CRIT_ALPHA (soil moisture control on photosynthesis) as being important controls on forest fraction, as the output seems to vary most with these parameters.

## 2.5 Training an emulator

The simulator FAMOUS, although relatively computationally cheap, is not fast enough to evaluate at every viable candidate point within input space, termed $\mathcal{X}$. We therefore use a computationally cheap statistical proxy to the simulator, called an emulator. The emulator provides a prediction of simulator output at any required untested input, many orders of magnitude faster than the original simulator. Once trained, any analysis that might have been done with the simulator can be done with the emulator, with the proviso that we must include an extra uncertainty term to account for the fact that the emulator is not a

perfect prediction of the simulator output. We use a gaussian process emulator that assumes zero uncertainty at points where the model has already been evaluated, growing larger away from those points, and dependent upon a set of hyperparameters that are trained at the same time as the emulator.

We treat the output $g(x)$ of the simulator FAMOUS as a deterministic function of a vector of input parameters $x$. The emulator is a nonlinear regression model conditioned on a sample, or ensemble, and provides a prediction of simulator output

and corresponding uncertainty.

We build a number of emulators of the ensemble, the details for each depending on the application. All use the DiceKriging package (Roustant et al., 2012), in the statistical programming environment R (R Core Team, 2016).



DiceKriging allows the user flexibility in specifying the emulator, and then estimates parameters of the statistical model using the training data. We verify the quality of the emulators, using a leave-one-out cross validation metric, ensuring that the accuracy and uncertainty estimates of the emulator are consistent across the ensemble (see supplementary material Fig. S2).

## 2.6 History matching

We aim to repeat the achievement of Williamson et al. (2014) to use history matching to find a region of parameter space that is consistent with observations to within the level of observational and acceptable simulator uncertainty. In practice this means finding a set of input parameters where the output of the model is deemed tolerably close to the observations, given uncertainty in the observations and known deficiencies of the model. Constraining parameters in this way should help identify the range of projected futures of the forest that are consistent with the observations, rather than a single set of "best" parameters.

A key distinction from the practice of model calibration is that the set of statistically consistent inputs are not accepted, but instead are deemed "Not Ruled Out Yet" (NROY). As such, we regard them as conditionally accepted, contingent on new observations or information. History matching was developed by Craig et al. (1997), and has been used extensively in hydrocarbon extraction sciences, and astronomy (e.g. Vernon et al. (2010)). Sometimes termed precalibration, It has been used to confront climate simulators with observations, for example by Lee et al. (2016); Williamson et al. (2013); Holden
et al. (2009). McNeall et al. (2013) investigated the potential of an observational dataset to constrain input space using history matching.

Observations of the system are denoted $z$, and we assume that they are made with uncorrelated and independent errors $\epsilon$ such that $z = y + \epsilon$, where $y$ represents the true state of the climate being observed. If we denote the "best" possible set of input parameters $x^*$, and assume that the simulator contains a systematic structural error $\delta$, then the observations can be related to
the input parameters

$$z = g(x^*) + \delta + \epsilon. \tag{1}$$

If the simulator were fast enough to evaluate at a large number of candidate points for $x^*$, this region could be found by standard Monte Carlo or optimisation methods. Our simulator FAMOUS, although relatively computationally cheap, is not fast enough for this. It is also our intention to develop methods that can be used on even more computationally expensive
simulators. We therefore again use the emulator as an efficient proxy for the model output, replacing $g(x)$ with $\eta(x)$ in Eq. (1), and including a term for emulator uncertainty in the history matching calculations.

Each point in input space is assigned an Implausibility $I$, according to Eq. (2). The forest fraction at a sample of points in input space are calculated, along with uncertainties, using the emulator described above. Inputs that produce forest fraction that is further from the observations are deemed more implausible. Those same inputs are less implausible if there is uncertainty
about the observation, about the model discrepancy, or the emulated output at that input:

$$I^2(x) = |z - E[\eta(x)]|^2 / [\text{Var}(\eta(x)) + \text{Var}(\delta) + \text{Var}(\epsilon)]. \tag{2}$$



A threshold of implausibility, above which a candidate input can be safely ruled out as implausible, is usually set to 3; roughly equivalent to a 95% credible interval of a posterior distribution, if using a Bayesian analysis. This is due to Pukelsheim's three-sigma rule; that for any unimodal distribution, 95% of the probability mass will be within 3 standard deviations of the mean (Pukelsheim, 1994).

5    Any input parameter set that has an implausibility score below the threshold is designated "Not Ruled Out Yet" (NROY), and is retained for further analysis. It should be noted that this does not imply that the input setting is *good* merely that the evidence from observations is not sufficient to rule it out as implausible: this may change as more observations, or more simulator runs become available.

## 3    Analyses and Results

10  ### 3.1    An initial history match

In this section we find regions of land surface parameter space in FAMOUS that remain NROY given some defensible assumptions about observational uncertainty. Figure 3 shows how the regionally aggregated simulated forest fraction varies across the ensemble. The figure shows histograms of the number of ensemble members with a particular forest fraction, compared with the corresponding observations. Although the simulator was not run at the "standard" or "default" parameter settings in the ensemble, we can use the emulator to estimate its output and uncertainty ($\pm 1$ standard deviation) at those settings, and show these on the plot, in black.

The model run at the standard inputs underestimates the forest fraction in the Amazon region by a considerable margin (a best estimate of more than 0.3). The other tropical forests are slightly overestimated, while the North American forests are very slightly underestimated. Global forest fraction is simulated very near to the observed fraction. Most of the ensemble members overestimate forest fraction in Central Africa, Southeast Asia, and North America. Some ensemble members simulate an Amazon forest fraction around, and indeed above, the observed fraction. This gives us cause to hope that it is possible to find a set of parameters where the Amazon and other forests are simultaneously well simulated, without using a simulator discrepancy function.

A target for a history matching exercise is to find regions of parameter space where simulator error is removed, or minimised to a level consistent with observational uncertainty. In practice, this requires finding a region where the large negative bias in Amazon forest fraction is minimised while keeping the other forests well represented.

We allow an observational uncertainty of 0.05 (one standard deviation) in each of the Amazon, Central African, South East Asian and North American forests. This corresponds to an expectation that the true 95% confidence interval of $\pm 0.15$, following Pukelsheim's rule. This range is nearly a third of the available range of zero to one, and we contend that it would be hard to argue that this is an over-constraint.

We sample from the emulator uniformly across input parameter space, history match using all four individual forest observations, and visualise the space where $max[I] < 3$. Figure 4 shows a density pairs plot of the approximately 12% of the 10,000 samples from the emulator that are Not Ruled Out Yet by the history match.



**Table 2.** Implausibility $I$ of forest observations at default input parameter setting of FAMOUS

| Observation | Implausibility $I$ at default parameters |
|---|---|
| Amazon | 3.99 |
| Central Africa | 0.56 |
| Southeast Asia | 1.24 |
| North America | 0.27 |

Does this region represent a viable set of inputs, perhaps to replace the default set of parameters? Where it appears that we may have found regions where both Amazon and other forests are plausible, we are suspicious of this region, for three reasons. First, the default set of parameters is ruled out, in this case by comparison of the simulator with observations of the Amazon (Table 2).

Second, it appears that in the active parameter space projections, these candidates are near the edges and corners of the input space considered plausible. The failure to rule out these points could be due to a relatively large emulator uncertainty. When parameters near the edge of an experimental design are suggested as NROY by a simulator-data comparison, this can suggest an undiagnosed fundamental simulator discrepancy. Third, we plot the histograms of the "best estimate" emulator output at these NROY points (Fig. 5), we see that they can be seen as *compromise candidates*. In general, if the simulator is run at points

in this region, it will overestimate the Central African, South East Asian and, most likely, North American forest fraction while underestimating the Amazon forest fraction. They are still included as NROY at these values because of the combination of the emulator uncertainty and the assumed observational uncertainty.

In the remainder of this section, we use a number of analysis techniques to investigate why a region on the edge of parameter space initially considered plausible, that does not contain the default parameter settings, is identified as NROY.

## 3.2    Finding the active parameters with sensitivity analysis

We perform a sensitivity analysis to identify the active subspace of model inputs and quantify relationships between the model inputs and outputs. In a descriptive sensitivity analysis, we show emulated mean regional and global forest fraction with inputs sampled from across input parameter space in a one-factor-at-a-time fashion, holding all but one parameter at their standard values while varying the remaining parameter (Fig. 6). The emulator is not a perfect representation of the simulator, and so we

include the emulator uncertainty estimates at $\pm$ one standard deviation, shown as shaded regions in the plot.

V_CRIT_ALPHA, and NL0 are the most influential individual parameters when considered across the entire ensemble, and counter each other when both raised. The Q10 parameter has little or no influence on forest fraction. The TUPP parameter is important only to the Central African (termed "Congo" here, for brevity) and Southeast Asian forest fraction, much less important to the Amazon, and not important at all to the North American forests.



**Table 3.** Total effect sum in sensitivity analysis.

| Parameter | Total effect sum |
| --- | --- |
| V_CRIT_ALPHA | 2.03 |
| NL0 | 1.09 |
| LAI_MIN | 0.53 |
| F0 | 0.41 |
| TUPP | 0.25 |
| R_GROW | 0.13 |
| Q10 | 0.04 |

The relationships change across parameter space and are therefore dependent on the somewhat arbitrary range of the initial input parameters of the ensemble design. Sensitivity can change in importance as parts of input space are ruled out. For example, the forests are most sensitive to NL0 in the lower part of the ensemble range, and most sensitive to V_CRIT_ALPHA in the upper part of the ensemble range.

Following (Carslaw et al., 2013), we quantify the sensitivity of the simulated forest fraction to the input parameters, using the FAST methodology (Saltelli et al., 1999), as coded in the R package *Sensitivity* (Pujol et al., 2015). We calculate the global sensitivity of the model output due to each input, as both a main effect and total, including interaction terms (Fig. 7). V_CRIT_ALPHA (soil moisture photosynthesis control parameter) is the most important parameter across the tropical forests and globally, with a total effect index of around 0.6. In tropical forests, NL0 (leaf nitrogen parameter) is next most important, with an effect index between 0.2 and 0.3. In all cases, interaction terms are relatively unimportant, accounting for only a few percent of the variance. North American forests show slightly different results, with NL0 being the most important parameter with a sensitivity index near 0.4 followed by LAI_MIN (leaf area index parameter), at around 0.3 and V_CRIT_ALPHA at 0.25. This difference is unsurprising, as the North American forests are a mix of broadleaf and needleleaf trees, which will have different sensitivities from a broadleaf tropical forest.

Parameter Q10 has almost no influence on forest fraction, in line with expectation from land surface modellers. The non-zero estimate of sensitivity here is very likely due to the fact that the emulator is not a perfect representation of the simulator, and a zero sensitivity is well within the uncertainty bounds of the sensitivity analysis. Parameters TUPP and R_GROW have very little impact on forest fraction. Parameter F0 has virtually no influence away from the tropics, conversely, LAI_MIN is only important in the North American forest. If we sum the total effects (first order plus interactions) for all of the forests types excluding global (as it is largely made up of our forests), we obtain a rank for each parameter (table 3).

## 3.3 Mapping simulator error in parameter space

In this section, we examine the ability of the simulator to reproduce the observed forest fraction, how that ability varies across input parameter space, and assess the region of parameter space which is consistent with each of the forest fraction observations.





We show a map of simulator error in the the two dimensional space of the most important parameters identified in Sect. 3.2 parameter space, in Fig. 8. We sample the emulator across all parameter space, and plot the mean predicted difference between model output and the observations for each point. The maps appear noisy because of the impact of randomly chosen values of the remaining dimensions, but the structure is clear. For the Central African, Southeast Asian and North Amer-

ican forests there is a broad sweep of parameter space, running from low NL0, low V_CRIT_ALPHA to high NL0, high V_CRIT_ALPHA, where simulator error is close to zero. The Amazon input space does not have this region - only the high NL0, high V_CRIT_ALPHA corner has a simulator error close to zero, suggesting bias in the model that is not common to all of the forests. It is possible to find a portion of parameter space where the bias is similar for all simulator outputs in the low NL0, high V_CRIT_ALPHA corner. However, the bias is rather large (at least -0.6) at this point.

## 3.4   How much input space is ruled out by combinations of observations?

Williamson et al. (2014) discuss treating the model discrepancy as a "tolerance to error". We take this approach, and find the potential of the history matching technique to rule out parameter space under a number of scenarios of observational and tolerance to model structural error. The denominator of Eq. (2) is the sum of the squared variances of the emulator, discrepancy, and observational uncertainty. Our emulator uncertainty is set, but we can experiment with the overall uncertainty budget by

partitioning uncertainty between observations and model discrepancy at will.

Different observations rule out different parts of parameter space, while combining observations can be a powerful method of ruling out large parts of parameter space. A number of approaches to combining data in history matching are discussed in Vernon et al. (2010) and Williamson et al. (2013). A simple strategy is to calculate $max[I]$ at a candidate input across all data independently, and reject those candidates with a value larger than 3 in any. A danger of history matching using $max[I]$ is that

a single poorly specified emulator or model discrepancy term could lead to large swathes of parameter space being incorrectly ruled out. As the number of comparisons with data goes up, so does the probability of including a poorly specified model discrepancy. For example, comparing a model with a serious but undiagnosed bias could lead to all a priori plausible parameter space being ruled out as a poor match to the observations. For that reason, it is important to first combine knowledge and judgement about the system being modelled, and the way that the parameters represent their real world counterparts (or don't),

before simply relying on observations to remove plausible parameter space.

A conservative measure is to reject a candidate point only if it is judged implausible using a number of measures. This will tend to be more robust to a poorly specified model discrepancy term. Vernon et al. (2010) use the 2nd and 3rd highest implausibility score, where a simulator has implausibility scores for multiple outputs calculated. This is to guard against poor emulators, but in practice works just as well for poorly specified model discrepancy. An alternative suggested by Vernon et al.

(2010) is to use a multivariate measure of implausibility.

To understand the value of individual observations, we ask *what is our tolerance to error?* What level of uncertainty in observations or model discrepancy (or both) can we tolerate before our observations become ineffective for history matching? Figure 9 shows the declining proportion of input parameter space ruled out as we increase our tolerance to error, in a number of scenarios. Coloured lines indicate use of the individual, and combinations of, the forest fraction observations. Tolerance



to error is specified as a single standard deviation, so in practice, the full distribution of the uncertainty of the observation or discrepancy (e.g. the 95% range) will be three times as large, using Pukelsheim's rule.

North American, South East Asian and Central African forest observations constrain parameter space to between 40% and 50% of parameter space, even when our tolerance to error is very low. The proportion of NROY space increases quickly,

particularly using North American forest fraction, which becomes no constraint at all when our error tolerance is above 0.07 (1 standard deviation). The other forests offer some constraint up to about 0.1 (1 standard deviation), and the Amazon is more of a constraint, only completely losing power as a constraint when the standard deviation of our tolerance to error is above 0.15 (1 standard deviation).

Combining data, and using the maximum Implausibility of any dataset improves the constraint considerably, particularly

when the tolerance to error is low. However, we urge caution. The fact that a) the performance of the Amazon data set appears quite different from the other observations, and b) that all of parameter space is ruled out at lower values, even though there is emulator uncertainty, again raises concerns of a poorly specified Amazon model discrepancy.

An alternative and perhaps more robust calculation of tolerance to error can therefore be found by excluding the Amazon observations and using the maximum implausibility from the other observations. This excludes more input parameter space

than any single observation on its own, up to a tolerance to error of around 0.85 (1 standard deviation), where it performs in a similar manner to using Southeast Asian forest fraction.

To what extent do the input spaces that are NROY when history matching with two forests overlap? We suppose that data that suggest highly overlapping input spaces give us confidence that those input spaces are valid. Another perspective is that overlapping input spaces give us little extra information, and we should seek out those that minimise overlap. We sample

uniformly from the input space, and test each point using a comparison with each forest observation to see if it is ruled out or not. If a point has the same status using both forests in the history match, we class that as an overlapping point. Table 4 gives the proportion of the samples which have the same status using each permutation of two forests for the history matching.

The most similar input space is found if we use the Southeast Asian and Central African rainforests. Comparing these forests with the North American forests gives a fairly high overlap - 61% and 66% for Southeast Asia and Central Africa respectively.

The Amazon has markedly lower overlap with the other forests - 40% at the most with North America, and only 26% with South East Asia.

### 3.5   What do the individual forests tell us about the best parameters?

In order to more fully explore the causes of model discrepancy and its consequences, we make the illustrative assumption that that model discrepancy uncertainty is zero, and that observational uncertainty is very low. We sample a large number of points

uniformly across input space, assume zero model discrepancy uncertainty of zero and an observational uncertainty of 0.01.

We keep as NROY only those emulated samples where the implausibility (or maximum implausibility in the case of combined data) is below 3. Setting such a demanding threshold allows us to find and describe the relatively small regions in input space where the model performs best, in two cases. First, using the South East Asian, Central Africa and North American forest fraction in the history matching exercise, second using the Amazon forest fraction.



**Table 4.** Amount of overlap in NROY input space for forest combinations.

| Forest A | Forest B | Input agreement (%) |
|---|---|---|
| Amazon | Southeast Asia | 26 |
| Amazon | Central Africa | 33 |
| Amazon | North America | 40 |
| Southeast Asia | Central Africa | 84 |
| Southeast Asia | North America | 61 |
| Central Africa | North America | 66 |

Plotted in two-dimensional projections in Fig. 10, we see that the "best" set of parameters as defined by matching to the observed Amazon forest fraction, and to the other forests, form almost non-overlapping sets in the most active subspace comprising V_CRIT_ALPHA and NL0. Again, we see a swathe of input parameter space, running from low V_CRIT_ALPHA, low NL0 through high values of those parameters. This pattern is confirmed when using the individual data sets for history

matching (not shown). The three non-Amazonian forests have a high degree of overlap of NROY space.

When run at a single parameter set, FAMOUS struggles to simulate both the Amazon and the other forests simultaneously, at any parameter combination when using a low threshold of implausibility. The implication of this is that it is very difficult to reconcile the model simulation of the Amazon simultaneously with the other forests if there is little uncertainty about the observations. A model discrepancy term and corresponding uncertainty of some form is necessary to attain an adequately

performing simulator.

The emulator offers the advantage of flexibility, and we can predict the implausibility at any point in parameter space, identifying regions of input space where the model output is inconsistent with the observations. For example, in Fig. 11, two parameters are varied across the full ensemble range, while all other parameters are held at their default value. The green point marks the default input value, projected into the two-dimensional space. For this illustrative example, we use a "tolerance to

error" of 0.1 (1 standard deviation), which is the assumed sum of observation and discrepancy uncertainty.

Using the Central African (CONGO for brevity) rainforest to estimate implausibility of each point in parameter space, we see that the standard inputs are located in a deep "valley" of low implausibility. Generally, the implausibility is very low at the standard settings. There are regions where implausibility may be equally low or lower, existing as planes within the multidimensional space. However, there appears to be no evidence that the standard set is implausible, given this data.

In contrast, using the Amazon as an observation, the shape of the plausible regions seems very different when projected into this two dimensional space. There are no longer valleys of NROY space, but a larger region that appears off to one side of the design input space. In addition, the standard values are often close to or at the boundaries of implausible space.





## 3.6 The forests at best parameters

To examine the implications of using each observation separately to tune the model, we use the emulator to project the forest at the set of "best" inputs for the alternative forests. We find input parameters where the model reproduces each forest, with a very small tolerance of error. We then use the emulator to project the Amazon forest fraction using the "best" parameters for each forest, and the forest fraction for each of those forests using the "best" parameters for the Amazon in Fig. 12. As there is some uncertainty, due to emulator uncertainty and a small tolerance to error, these are plotted as histograms.

We find that the using the best set of parameters as defined for each non-Amazon forest would most likely lead to an underestimate of the Amazon forest fraction by around 50%, compared to the observed fraction (around 0.3, compared to an observation of around 0.6). Conversely, using the best parameters as defined for the Amazon leads to an overestimate of the other forests - around 0.3 for the tropical forests, and 0.15 for the North American forest. This occurs even even though the observed aggregate forest fraction is very similar for the tropical forests.

To further explore this difference, we project the "best" set of input parameters, found using the Amazon and African forest to match the simulator against, over a map of the entire FAMOUS land surface. In each case, an independent emulator is trained on the ensemble for each grid box. The maps of the mean forest fraction for each parameter set, and the difference between them, is shown in Fig. 13.

We see that even using the "best" Amazon parameters, the simulator underestimates the Amazon coverage in the North East of South America. This makes it very difficult to approach a sensible forest fraction, even when boosting the forest fraction in places where the model does have forest cover.

## 3.7 History matching allowing for discrepancy in the Amazon

Taken together, the analysis in the previous sections show that the inputs where FAMOUS best simulates Central African, South East Asian and North American forests cover a similar input space, whereas the best inputs for the Amazon are in a different region. This suggests that we should use a non-zero-mean discrepancy for the Amazon: allowing the Amazon to be less vigorous in our simulations, while maintaining that the simulator output should broadly match the other forests. We do not have enough information to create a more detailed discrepancy function: for example, one that varies across parameter space.

We perform a history match using all of the forest observations, along with a simulator discrepancy term for the Amazon forest. We use the best estimate of the difference between Amazon observations, and that simulated by FAMOUS at the default set of parameters as the best estimate of the discrepancy mean. The difference in forest fraction at the default parameters is approximately 0.3. Figure 14 shows the histograms of NROY input space using this discrepancy term, along with credible estimates for observational uncertainty (1 standard deviation = 0.05) and tolerable discrepancy uncertainty (1 standard deviation = 0.03). The corresponding two-dimensional density plots of NROY emulated samples can be seen in Fig. 15. The remaining NROY input space represents around 57% of the original input space defined by the input design, meaning that we have ruled out 43% of the space. This contrasts with ruling out around 88% of the space in the inital history match in Sect. 3.1.





Finally, marginal histograms of the relative density of NROY points for each individual input parameter are shown in Fig. 16. These indicate that no part of the marginal input space is completely ruled out, and so we cannot "constrain" any of the parameters in an individual dimension. However, the relative frequency of NROY points is higher in some locations than others - low in NL0 and high in V_CRIT_ALPHA for example, suggesting a higher probability that the best estimates of the parameters is in these regions.

## 4   Discussion

Uncertainty in carbon cycle and land surface process contributes significantly to uncertainty in future climate change. There are a large number of uncertain input parameters to carbon cycle and land surface components of climate models, and our study attempts to use comparisons of the model with observed data to constrain some of the key parameters. We find that forest fraction does not offer a marginal constraint on the parameters: that is, there is little or no constraint on each parameter individually, but there is a significant constraint on the joint input space of the parameters. Approximately 43% of a priori parameter space is ruled out, which is relatively little compared to other studies. This is explained by two factors: 1) our observational uncertainty is assumed conservatively large, and 2) we have only a single wave of history matching. A further experiment could run the climate model within the NROY space in order to reduce emulator uncertainty, and provide a basis to further rule out input space. The value of further waves of history matching might be diminished by the fact that the simulator likely has a large discrepancy in the Amazon, and the model discrepancy uncertainty is likely a large component of the overall uncertainty budget.

Our analysis illustrates the challenges in distinguishing between model discrepancy, parameter uncertainty and observational uncertainty during model development. For example, forest fraction in the model can be tuned largely by using the two most active parameters: V_CRIT_ALPHA and NL0. As these parameters alter forest fraction in counteracting directions, a number of solutions can be found that give plausible forest fractions. Information from outside sources about the "true" (or appropriate) values of one these parameters might therefore offer a strong constraint on the value of the other. NL0 is the leaf nitrogen parameter - the ratio of nitrogen to carbon found in leaves. In theory, this is something that is well observed and recorded, but it is uncertain what value should be to reflect the observational range across the spatial scale of FAMOUS. Nitrogen content determines the maximum photosynthesis, and therefore how much $CO_2$ can be assimilated, or the productivity of a plant. Low (high) NL0 values correspond to low (high) nitrogen content, and hence a low (high) productivity plant. V_CRIT_ALPHA is the soil moisture threshold below which plants are water limited. So if this parameter is high, then the plant is more often in a water limited regime. Where as if it is low, then a plant is not as often water limited.

If we use observations of the Amazon rainforest, along with the other forests major forests in the history matching exercise, we find that we can rule out a large swathe of parameter space, including the standard set of parameters. There appears to be a corner of parameter space that is NROY under these conditions. While it first appears that we have found a region of parameter space where the model output is tolerably close to the observations given a zero-mean discrepancy, there are good reasons to be suspicious of this region. The region is close to the edge of the ensemble in the active parameter subspace, so





that uncertainty may dominate the implausibility calculation. The region excludes the default set of parameters, which are chosen the result of multiple lines of evidence and scientific judgement. Further, the information obtained from using each of the four forests shows that the Central African, Southeast Asian and North American forests all indicate very similar, highly overlapping NROY regions. In contrast, the NROY region suggested by comparing FAMOUS to observations from the Amazon

is very different. Should we trust this region as NROY?

For illustration, we imagine a situation where we are forced to choose between keeping the default parameters and including a simulator discrepancy function, or rejecting them and accepting a candidate or candidates from the new NROY region. We argue that the fact that three of four data sets - and in different regions and types of forest - give us similar information about the parameters, and that they all include the default parameter settings as NROY, suggests that we should include a simulator

discrepancy function.

We urge caution with a naive or automatic application of history matching conclusions, particularly when using multiple observations for comparison with the simulator. Even in our relatively simple history matching exercise, there is a clear need to include model discrepancy, or increase model discrepancy uncertainty, or to apply a conservative version of the measure of implausibility. One strategy, adopted for example by Vernon et al. (2014) is to reject parameter space that has a second- or

third- highest implausibility metric larger than some threshold. This would be effective in the case of our comparison. Another strategy might be to reject only parameter space where the minimum implausibility is higher than some threshold. We believe that this would not rule out much input space in many circumstances. We call for more research on the behaviour of measures of implausibility, when the number of data comparisons is high, and there is a chance that many of them may suffer from structural biases.

We are able to offer a counter example to the hypothesis of Williamson et al. (2014), who found regions of parameter space where what was thought a structural error in the model was significantly reduced. In this case, we believe it likely that better observations would simply confirm that the "best" regions of parameter space for the Amazon and other forests were non-overlapping. While individual forest fraction observations may have some uncertainty, we would expect the uncertainty on the differences between those observations to be smaller. A systematic bias in the way that the forests are measured would be

common to all observations, for example, even though it would need to be taken into account in the uncertainty calculation for an individual observation.

## 4.1  Causes of discrepancy

What could cause this fundamental structural error in the Amazon? There are three possible causes - *external* and *internal* to the vegetation model, although a combination of these causes is not ruled out. First, is there a problem with the emulator that could cause such a bias? We believe that this is not the case, as the emulator performs sufficiently well across parameter space

in cross validation experiments (see supplementary material Fig. S2).

Second, is there a missing processes in the vegetation model, that impacts the Amazon in FAMOUS? It is possible that the real Amazon can access water to a deeper level than other forests, through deep rooting. This would cause a *low Amazon* bias, seen in the model output. If the simulated Amazon can't access water through deep enough roots, and model parameters were



tuned to make Amazon as vigorous as real world, other forests would be more vigorous in the model than in observations. A bias that leads to a reduction in Amazon forest extent (such as that climatological or root depth) is likely to lead to further rainfall reductions, and its associated warming, as the region loses water cycling capability that the forest canopy provided. This is a feedback, and can be expected to enhance any dry/warm bias that results from other factors, and in turn enhance any
forest loss.

Finally, does the model simulate the climatic boundary conditions of the forest well enough? Malhi et al. (2009) and Staver et al. (2011) note the dramatic influence of climate on Amazon forest cover, albeit mediated by fire and not included in FAMOUS. Evidence from previous studies shows that HadCM3, which FAMOUS is designed to replicate, does have some climatic biases in the Amazon. Cox et al. (2004) find that rainfall in the Amazon is underestimated, particularly along the North
East coastline. Precipitation is underestimated by approximately 20%. The dry season is too long (it starts a month early), and there is an underestimate of wet season rainfall. This precipitation anomaly persists in FAMOUS, although is perhaps not as severe as in HadCM3 (Jones et al. 2005, Fig. 4). Good et al. (2008) note that simulated Amazon dry season precipitation is closely tied to meridional sea surface temperature gradients in the region. Joetzjer et al. (2013) and Yin et al. (2012) note similar climatic biases across the CMIP5 archive. We suggest that attributing the simulator discrepancy to these causes might
be a fruitful direction for further study.

## 5   Conclusions

We analyse an ensemble of the fast climate model FAMOUS with the aim of constraining carbon cycle parameters through a comparison of simulator output with forest observations. We find that we are unable to constrain the parameters individually, but that areas of joint parameter space are effectively ruled out. With a defensible model discrepancy term for the Amazon, and
assumed observational uncertainty we are able to rule out 43% of the input parameter space defined by the ensemble design.

We identify moisture control on photosynthesis (V_CRIT_ALPHA) as the most important parameter control on forest fraction, with the next most important leaf nitrogen (NL0), parameter being approximately half as important, and that twice as important as any other parameter. These parameters have counteracting effects on the forest fraction, so we are unable to rule out a broad swathe of the joint space of these two parameters.

We suggest that we should exercise care if using observations of the Amazon rainforest to constrain the input parameters of FAMOUS, as an apparent structural bias in the climate model could lead to misleading results. Using the Amazon forest as an observational constraint suggests very different parts of input parameter space as *not implausible* than using other forests. Although we are able to find a region of parameter space that we are unable to rule out, given a defensible assumed observational uncertainty, we have reason to suspect that this region does not offer a credible alternative to default parameter settings. Further
investigation reveals that choosing the region would systematically overestimate the forest fraction of the Central African, South East Asian and North American forests, while simultaneously underestimating the Amazon. We fail to find a set of parameters that eliminates the discrepancy between the simulated fraction of the Amazon and other tropical and boreal forests. We suggest





that we cannot find a set of vegetation model parameters that improve the Amazon without making the other forests worse. This satisfies the criterion of Williamson et al. (2014) to identify a simulator bias.

Using a history matching technique, we investigate the limits of observational and model discrepancy uncertainty, beyond which observations no longer offer a constraint on input parameter space. We find that if this total error budget is larger

5 than approximately 0.1 (1 standard deviation of forest fraction), and excluding the Amazon rainforest as a comparison, the observations will not offer any form of constraint on the current ensemble, even in joint parameter space.

*Author contributions.* DM and all authors designed the analysis. DM conducted the analysis and wrote the paper. JW provided the FAMOUS ensemble and BB provided the observed forest fraction data.

*Acknowledgements.* This work was supported by the Joint UK DECC/Defra Met Office Hadley Centre Climate Programme (GA01101). DM

10 was supported on secondment to Exeter University by the Met Office Academic Partnership (MOAP) for part of the work.



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



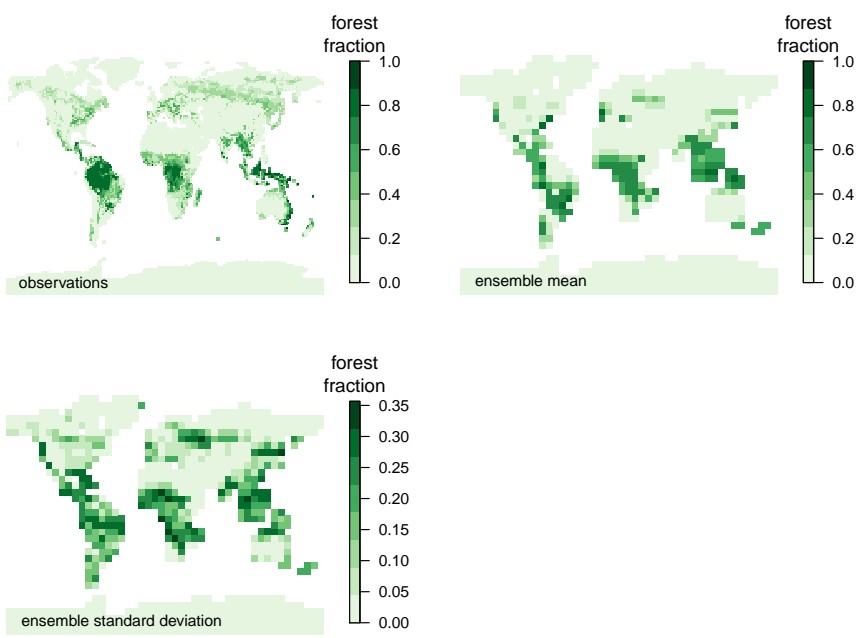

**Figure 1.** Observations of Broadleaf forest fraction (top left). Mean (top right) and standard deviation (bottom left) of broadleaf forest fraction across the 100 member ensemble of FAMOUS.





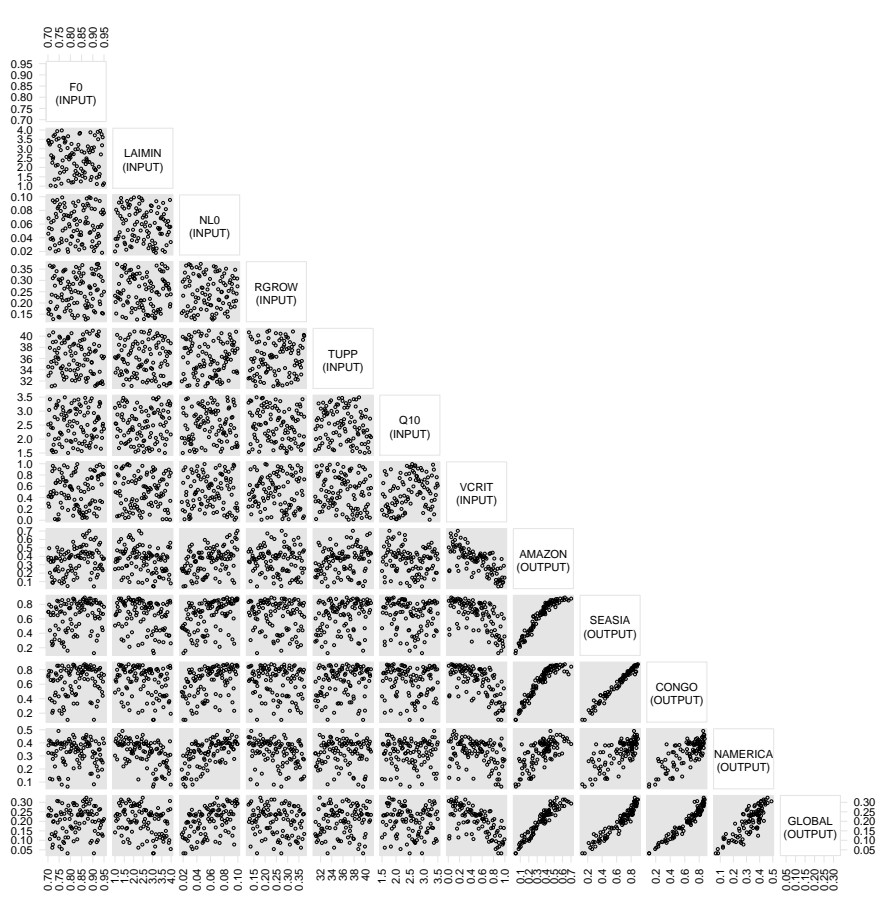

**Figure 2.** FAMOUS input parameters and forest fraction parameters, plotted against each other.



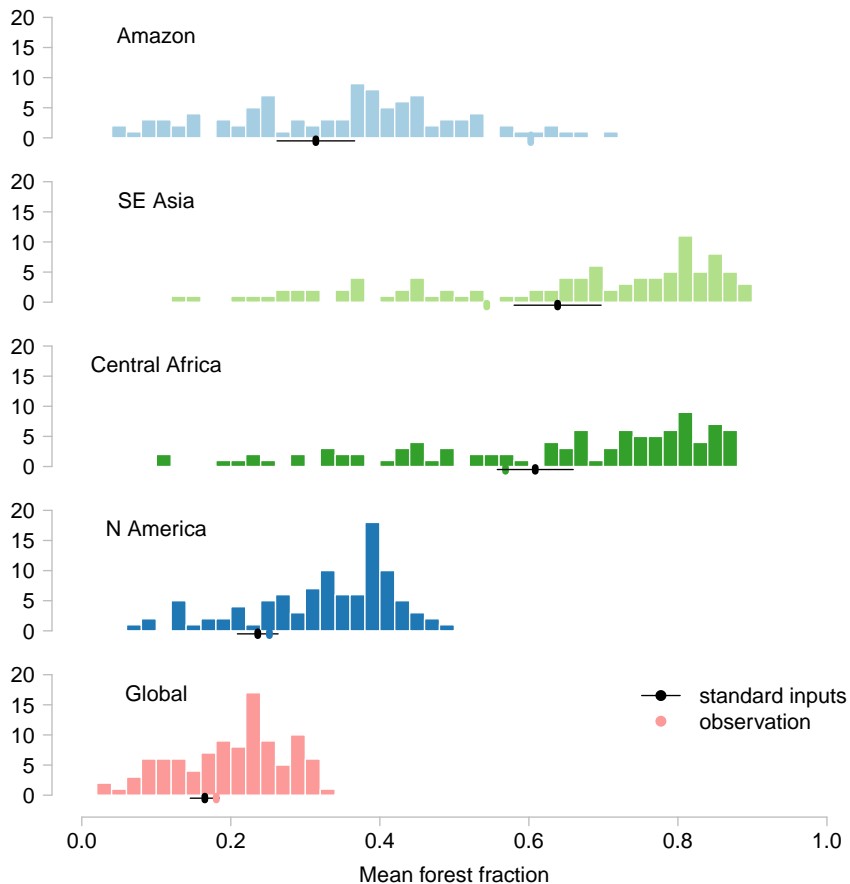

**Figure 3.** Histograms representing the number of ensemble members of a particular forest fraction in each region, and globally. Points plotted below the histograms represent the observed forest fraction (colours), and the forest fraction simulated at the "standard" parameters ±1 standard deviation (black).





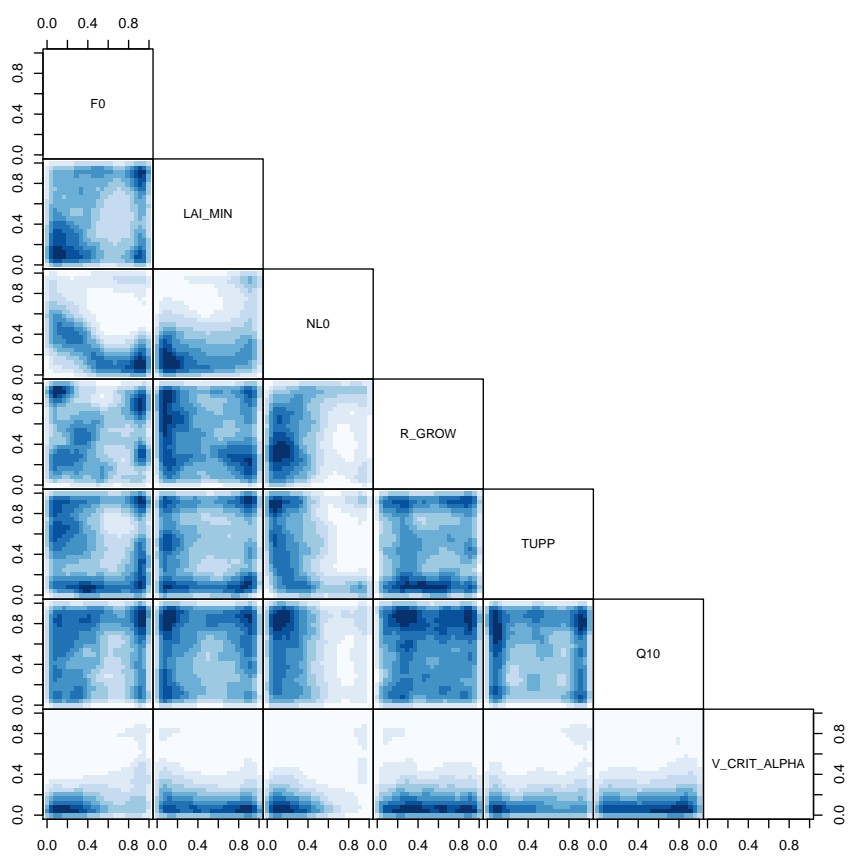

**Figure 4.** A density pairs plot of two dimensional projections of parameter space. The blue areas represent the density of NROY points, using all of the data, with an assumed observational uncertainty of 0.05 (1 standard deviation).




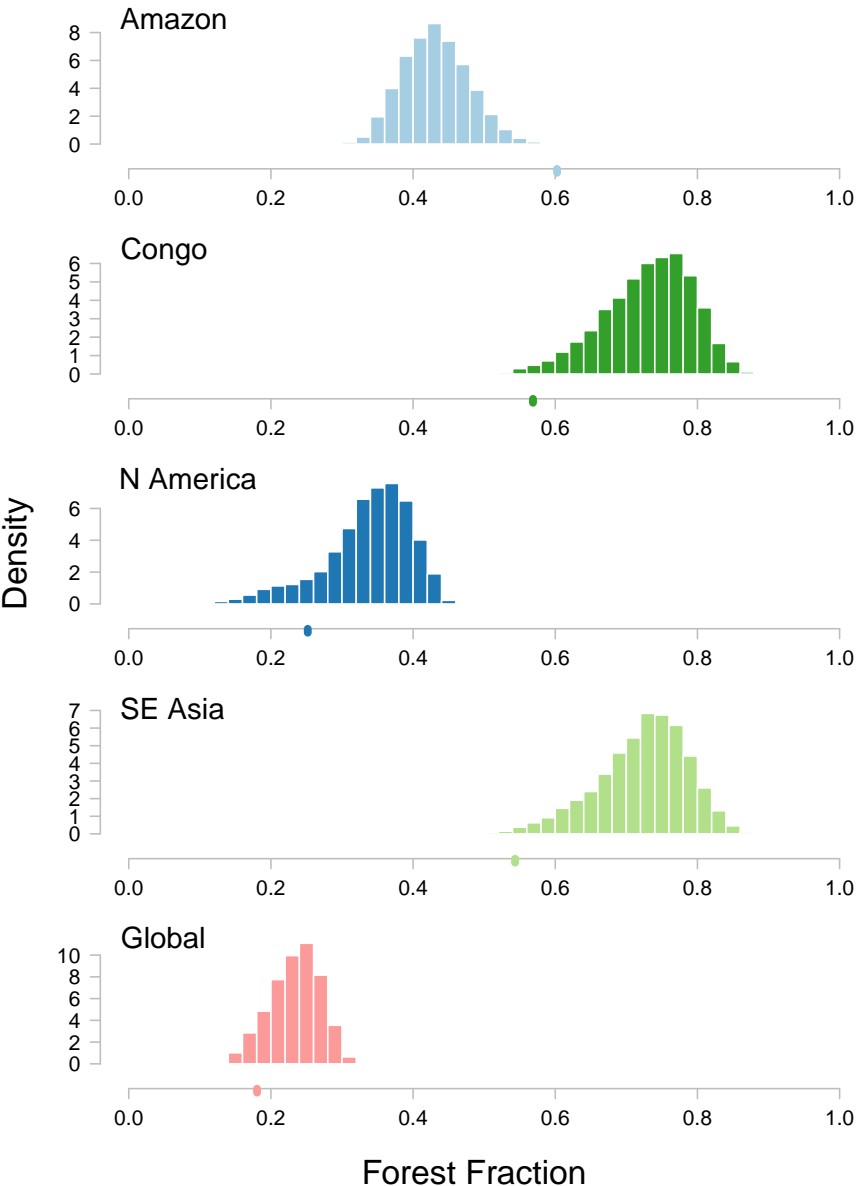

**Figure 5.** Best-estimate draws of forest fraction output from the emulator, at the set of points Not Ruled Out Yet when assuming a credible observational uncertainty. The value of the observed forest fractions are are plotted as a single point on the corresponding x-axes (a "rug plot").





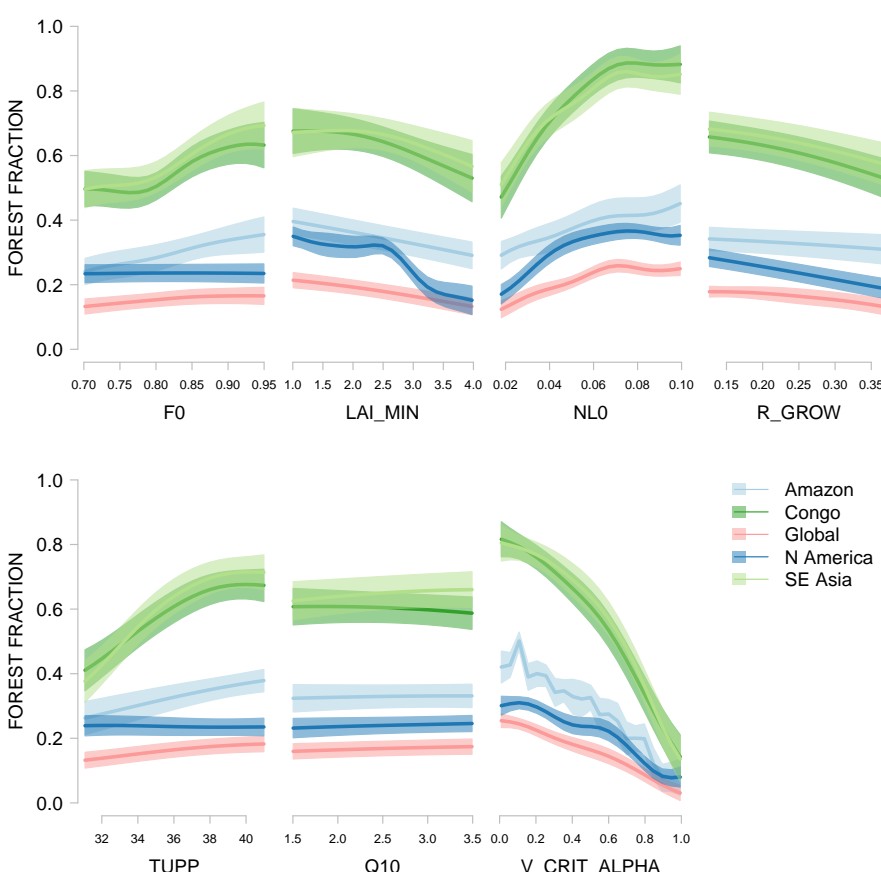

**Figure 6.** Marginal sensitivity of mean forest fraction to each input parameter in turn, with all other parameters held at standard values. Central lines represent the emulator mean, and shaded areas represent the estimate of emulator uncertainty, at the ±1 standard deviation level.



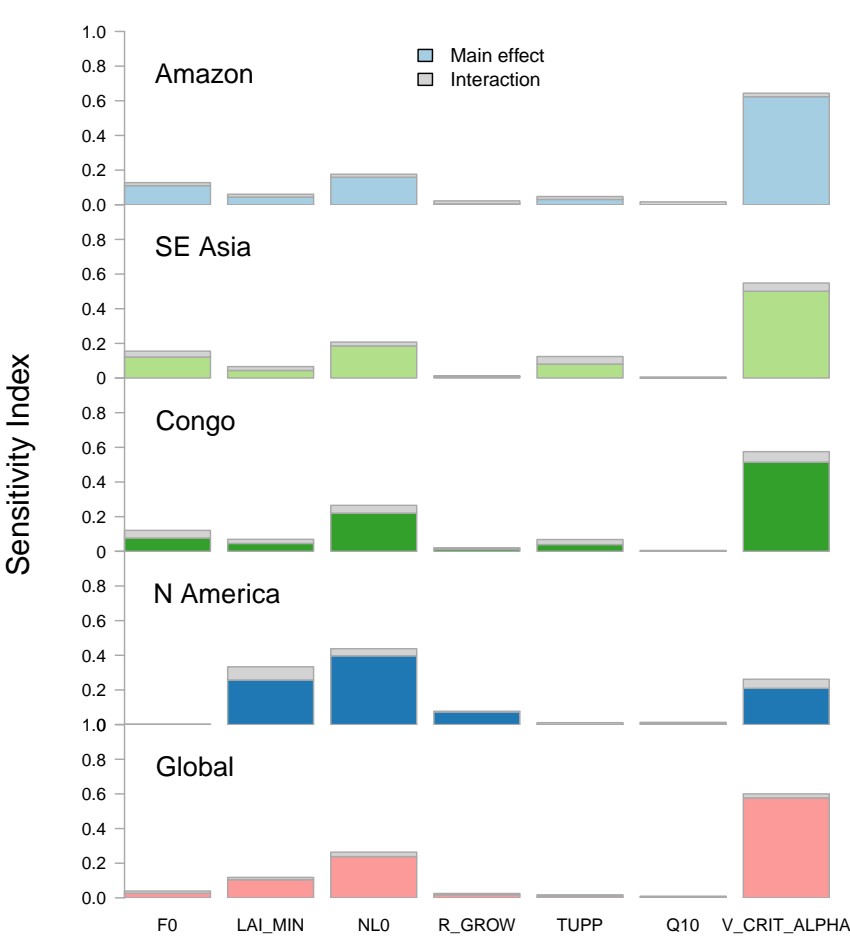

**Figure 7.** Sensitivity analysis of forest fraction via the FAST algorithm of Saltelli et al. (1999).





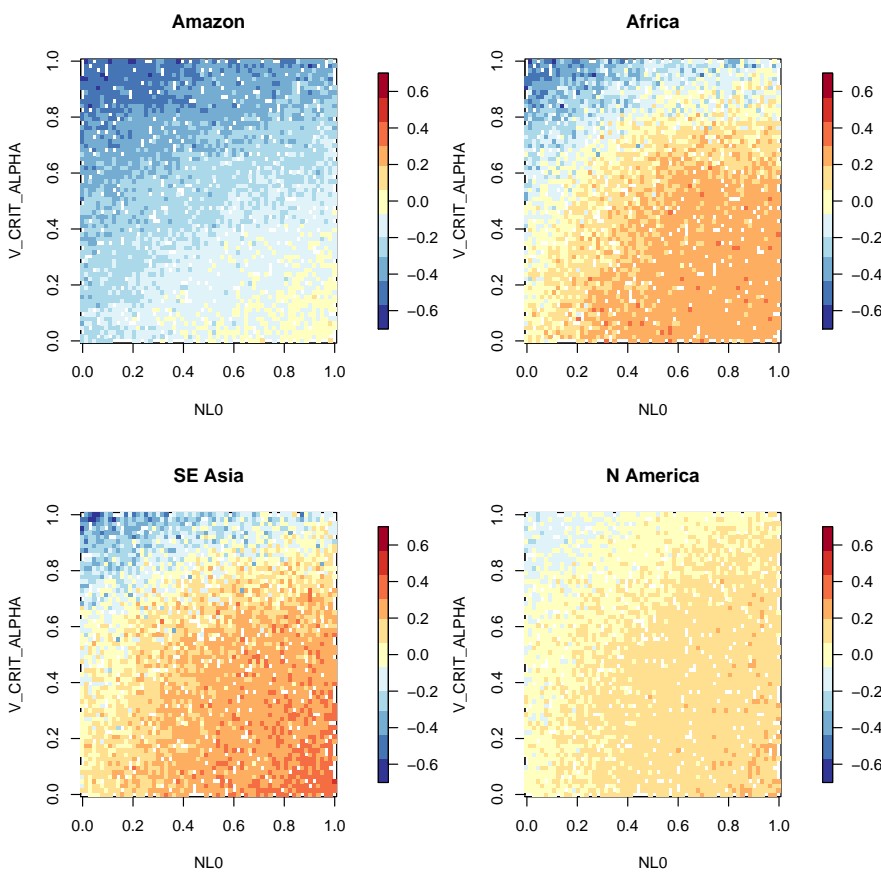

**Figure 8.** Maps of simulator error, in units of forest fraction, when projected into the two dimensional space of the most active parameters, NL0 and V_CRIT_ALPHA.





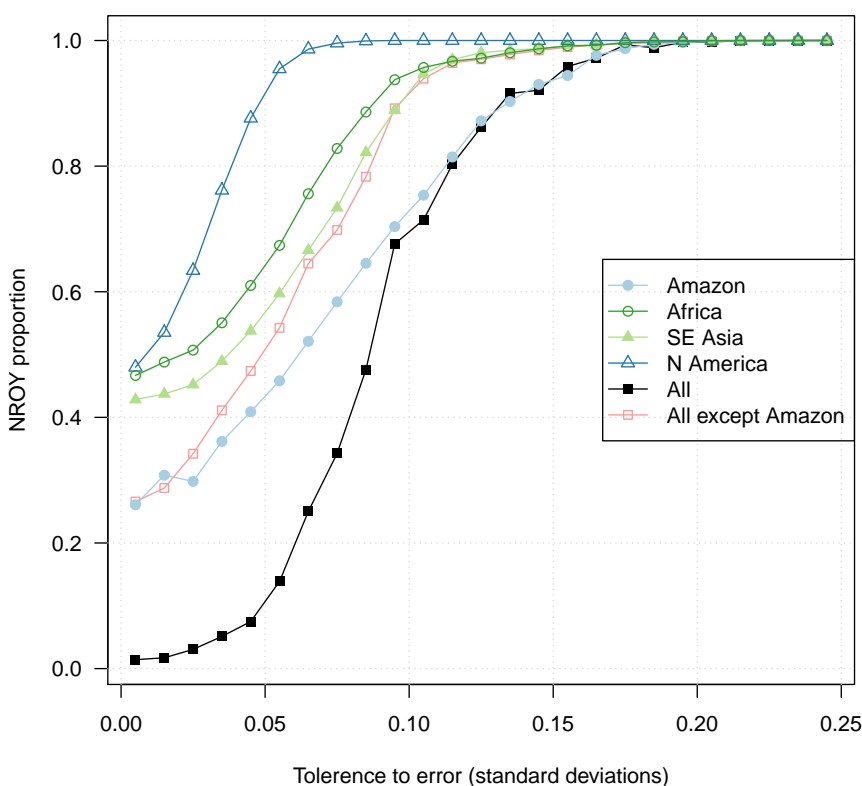

**Figure 9.** Proportion of NROY (Not Ruled Out Yet) input space plotted against "tolerence to error" - the total error budget including emulator, observational and simulator discrepancy uncertainty.





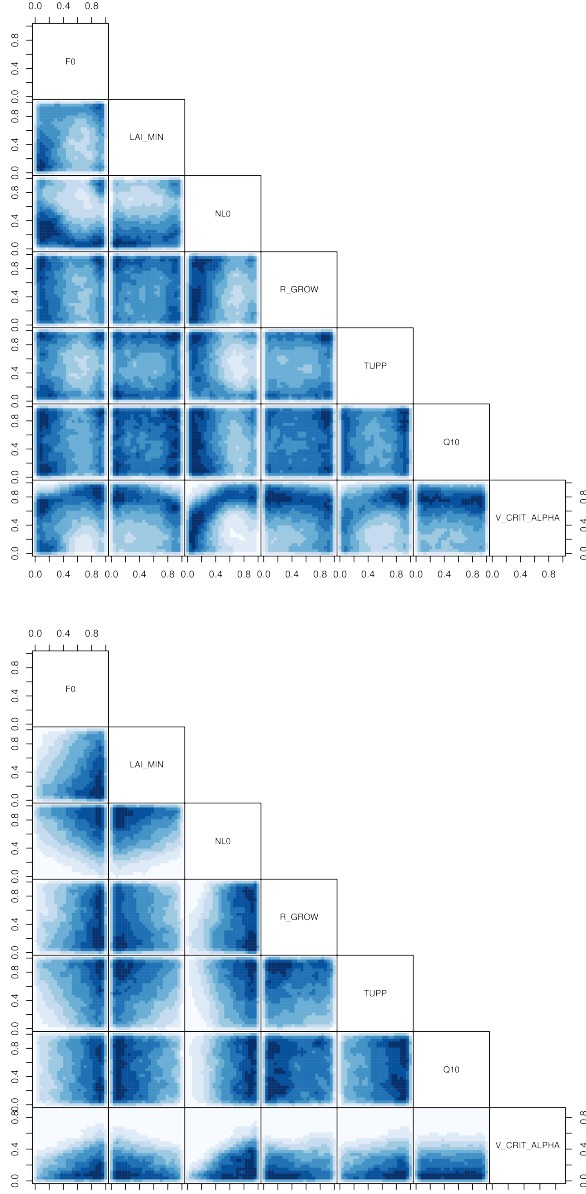

**Figure 10.** Marginal density of input parameter sets consistent with a very low "tolerance to error", and perfect observations, for the North American, Southeast Asian and Central African forests combined (top) and the Amazon (bottom). Dark blue regions indicate those with the highest concentration of NROY candidates, and therefore most compatible with the observations.



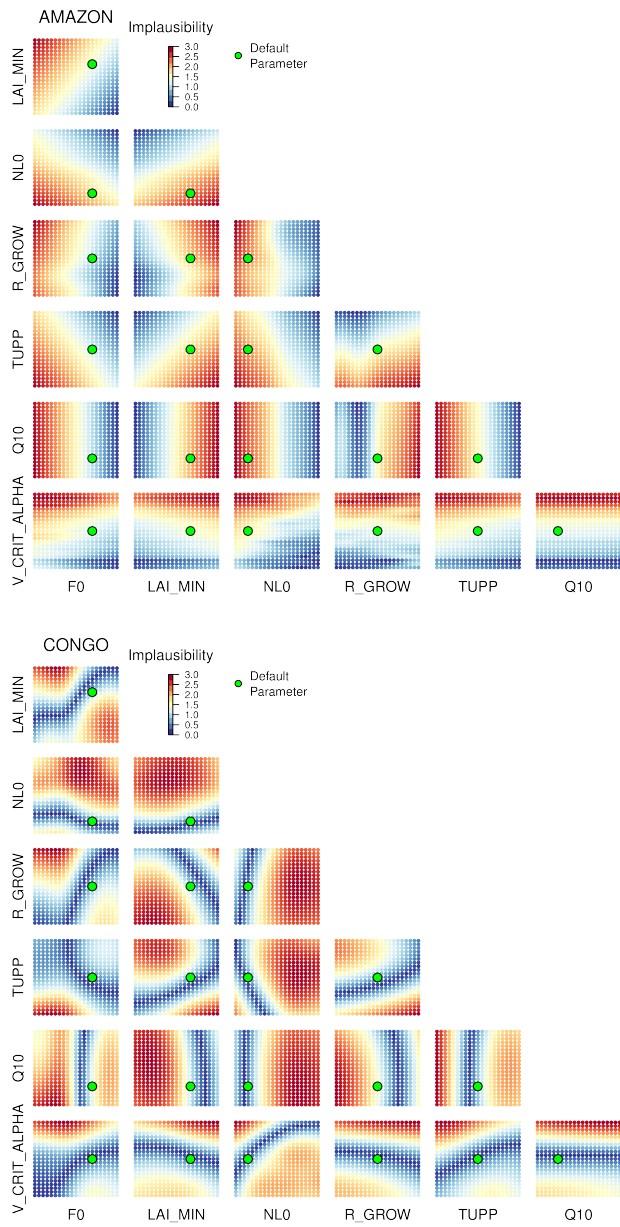

**Figure 11.** Implausibility, given a "tolerance to error" of 0.1, varying two parameters at a time and holding all others at their default values. Amazon forest (top) and Central African forest (bottom). Blue indicates regions where the model best simulates the individual option, while red indicates regions where the model simulates the forests more poorly. The green point indicates the location of the "standard" set of parameters for FAMOUS in the varied dimensions.





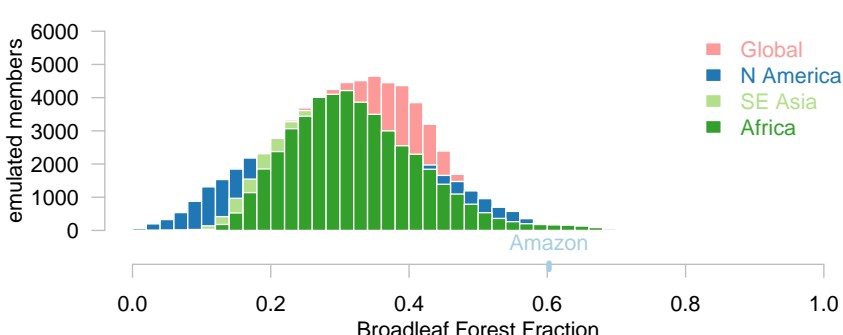

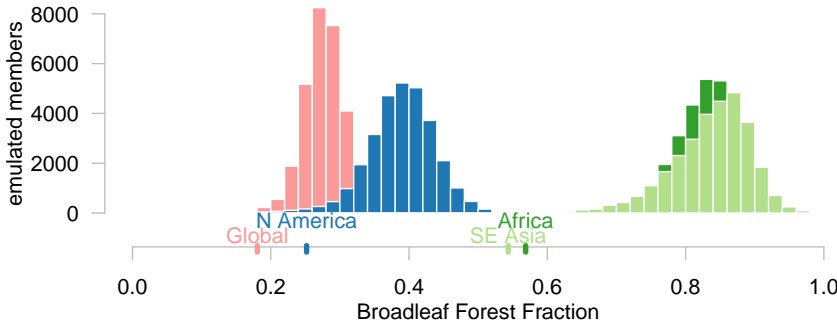

**Figure 12.** (Top) Forest fraction in the FAMOUS Amazon at the set of parameters where the FAMOUS best matches each of the other forest observations. (Bottom) Other forests in FAMOUS at the set where the FAMOUS Amazon best matches observations. Observed forest fractions are shown as marks underneath the histograms.





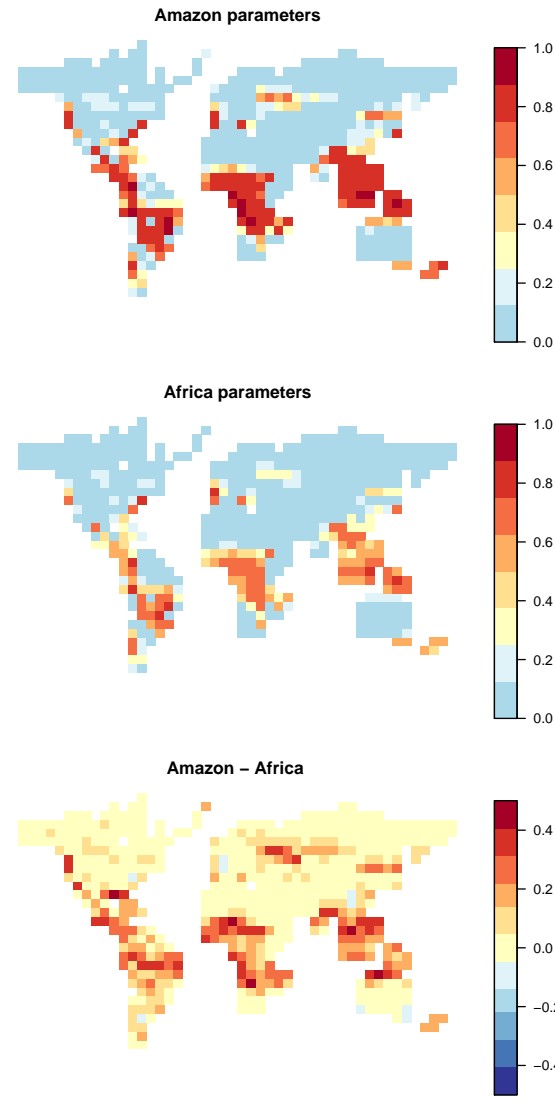

**Figure 13.** Maps of mean broadleaf forest fraction, over the "best" set of parameters found for the Amazon (top) and the Central African forest (centre). The difference between the two is mapped at the bottom.



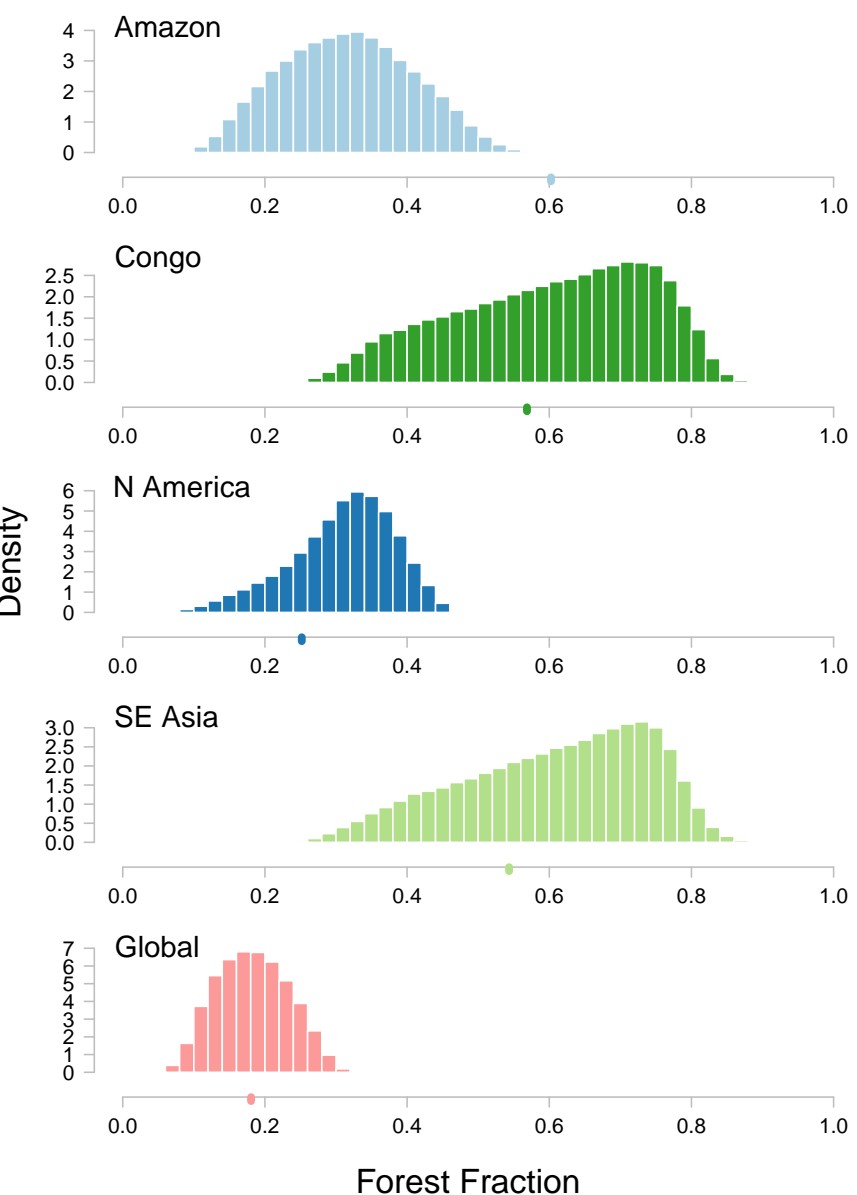

**Figure 14.** Histograms of emulated simulator output using credible estimates for observational uncertainty, a model discrepancy term for the Amazon, and credible discrepancy uncertainty.





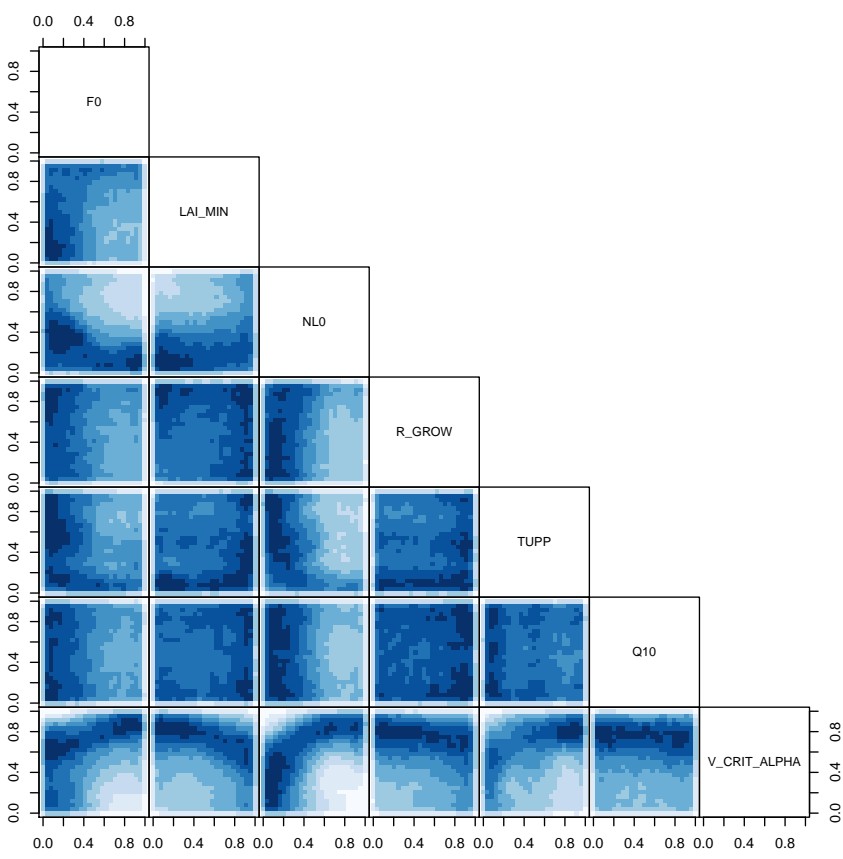

**Figure 15.** A density plot of the two dimensional projections of NROY samples from the design input space, using a all forest observations and a discrepancy function for the Amazon.





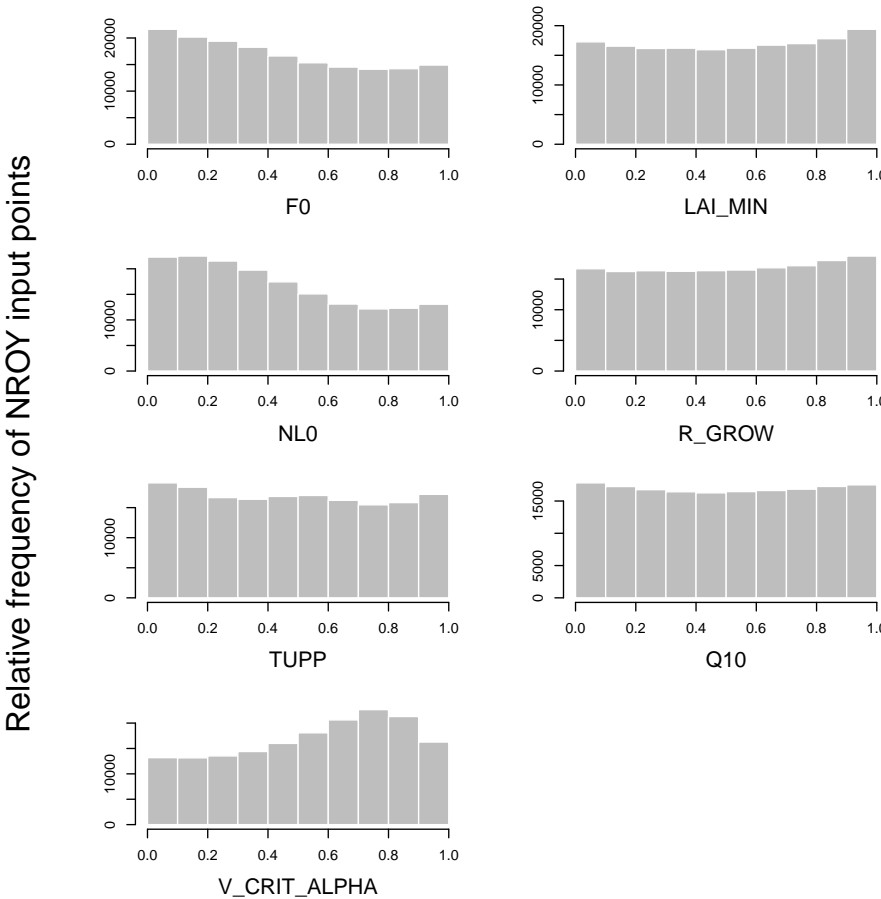

**Figure 16.** Marginal histograms of the relative frequency of NROY emulated input points in each dimension of parameter space, using all forest observations and a discrepancy function for the Amazon.