# Peer review of "Figure 1. A map of the forest regions used in the study. Regions are: Amazon $15^{\circ}\text{S} - 15^{\circ}\text{N}$ , $270^{\circ}\text{E} - 315^{\circ}\text{E}$ ; Central Africa; $15^{\circ}\text{S} - 10^{\circ}\text{N}$ , $7.5^{\circ}\text{E} - 30^{\circ}\text{E}$ ; SE Asia $12^{\circ}"

_Earth System Dynamics, 2016_

## Short Comment (SC1) · 31 May 2016

I don't believe this paper should be taken seriously, as it lacks any spiral graphics. Please see attachment for example of best practice in this area.

———————————————

[Figure]

[Figure]

**Fig. 1.**

---

## Short Comment (SC2) · 31 May 2016

Thanks to R. Campbell for the suggestion in using this cutting edge graphical technique. Unfortunately, the aggregated data used in this study effectively covers only a single time point, and so is impossible to adapt to the "viral spiral" plot. We look forward to using the suggested graphics technique in the future, should we extend the current study to use time series data.

---

## Referee Comment (RC1) · Anonymous Referee #1 · 20 Jun 2016

Review of paper: "The impact of structural error on parameter constraint in a climate model" by Doug McNeall et al.

Thank you for inviting me to review this paper.

The paper is interesting and important as it addresses whether a component of a GCM can be calibrated for one part of the globe, but applied elsewhere. Climate models are heavily dependent on transferability of parameterisation of sub-model structure, and a knowledge of when this fails is important.

I can see the aim of the paper, and it will be useful to have in the literature. However there did seem to be a slightly excessive use of statistical terminology. That's fine if the statistics is of standard form, but that's not the case here as the methods utilised are more novel. Please ensure that the literature is cited sufficiently well that any part of

this paper can be understood by calling upon the appropriate referenced papers.

Below are some comments that the authors might like to consider for a revised manuscript:

Overall points

The title is possibly too general. The emphasis is on DGVM modelling of forests, not general overall issues of structure.

The Abstract needs to be something that can be read in isolation, such that the reader can obtain a strong idea what the paper is about. To my mind, there is some repetition (e.g. three times says this uses "a history matching approach", and yet doesn't define what this actually is). Removing repetition can make space for more details. Extra description of the parameters changed would be helpful, rather than a vague "parameters that lead to a realistic forest fraction".

Reviewing this, I'm trying to really understand what the main thrust of this paper is about, in the statistical/algorithm sense. Can I confirm that the over-arching message is that quantity delta in Eqn (1) is important, can be characterised, and shows geographical variation. To my mind, that is a powerful result. It basically says if (i) not enough process representation is introduced in to a model, then structure deficiency gets masked by parameter fitting, and (ii) doing so will create problems between different locations. It would be nice to acknowledge that structural errors presumably also reduce confidence in any model for future projections, even when just at a single region where it performs well for contemporary periods.

Page 9, starting "Does this region represent......". This is a critical part of the paper, discussing how in effect a standard best-fit might not always be appropriate. Can the discussion be led back to Eqn (1), and in particular the structural delta parameter? (Also line 1, page 9, I cannot see in a Table or diagram what the alternative potential values are, for comparison against the default inputs – apologies if I've missed some-

thing). Where are the local, or continent-scale, delta values given?

Details

P2, line 10. Again, please give the reader some idea what "History matching" is, given other quantities such as "calibration" and "tuning" are defined at this point.

Around lines P2, lines 25-29. It would be really nice to have more concrete reasons why emulators, parameterisations etc are needed. This usually comes down to two factors: (1), computational speed prevents very high resolution modelling, even if the processes are more fully understood. For example, parameterisation of convection. (2), we don't know what the values should be, and these may exhibit strong regional heterogeneity. The latter is more the case for this paper, with questions asked as to what are the appropriate number of plant functional types that should be in land surface models – and if the number is high, can for example EO provide the values.

Check notation is consistent throughout. P3, line 23, FAMOUS is described as a "climate simulator". In the minds of the authors, is this different to a standard GCMs. Do they regard FAMOUS's reduced resolution as removing it from being regarded as a full GCM?

Again, in Section 1.3, this is now the 7th or 8th time that "history matching" is mentioned – it would be good to help the reader as to what it is, even if it is only to provide a methodological citation at this point.

Cox (2001) is a technical note. Better to give a peer-reviewed reference?

P5, line 1. I don't understand the context of the sentence: "The Amazon region is not wet enough for a fully humid region to exist". If this refers to the FAMOUS model, and in particular its atmospheric response, then this will make any DGVM fail if rainfall totals are too small.

P5, discussion of beta parameter. In a similar vain to the comment above, is it OK to treat the atmospheric beta parameter as a "nuisance" parameter? Isn't there a risk that

errors in GCM-projected precipitation – for example - will affect best-fit parameters in Table 1?

P5, line 18. From code that is shared with other centres, TRIFFID has a rapid spin-up option to near-equilibrium. Does it really need 10000 years?

Trivial thing, but it might be nice in Figure 2 to write as S.E.Asia (not SEASIA).

Can I confirm that a reader could find all details of the emulator in the Roustant et al 2012 paper. So, for instance, what a "leave-one-out cross validation metric" is.

Figure 7 I find very useful as it allows assessment of the geographical differences, providing more information that the global parameterisation Table 3. There are quite a few statistical methods available to determine parameter importance and/or nuisance parameters. An extra sentence stating what additional benefit the FAST algorithm brings would be helpful – i.e. beyond just the Saltelli reference.

Figure 8 is important as it shows how the Amazon has a difference response. Or put another way, a calibration of NL0 and V_CRIT_ALPHA for the Amazon could find a pair of parameters that would clearly be sub-optimal when applied to the other 3 regions. And vice-versa. I'd like to see more discussion around Figure 8, how it demonstrates the structural problems (i.e. very different responses to NL0 and V_CRIT_ALPHA, depending on location), and again – can this be related back to the delta parameter? This will also link better to the paper title, which is about model structural problems.

Figure 13 is nice and clear, and in many ways it is a shame that the paper is so long in technical details before getting to that point. Obviously this is a slightly naïve comment, but could it simply be that the trees of the Amazon have evolved differently to those of Africa. This could possibly be due to different imposed climatologies that the trees have adapted/acclimated to. So one conclusion of this paper could simply be that any land surface model such as TRIFFID requires a parameter mask, or ancillary fields, that are different for different places. The paper hints at this, page 16, in "Causes of discrepancy", where different rooting depths are considered. One future work extension might therefore be to include a root depth as a geographically-varying parameter, to add to those in Table 1? Would this then collapse delta down to zero for all locations?

---

## Referee Comment (RC2) · R. D. Wilkinson (Referee) · 22 Jun 2016

This paper describes a thorough and detailed investigation into the ability of FAMOUS to predict forest fraction. The paper starts from the pretext of being given an ensemble of pre-run simulator evaluations and observation data corresponding to some of the outputs, and being asked to estimate some of the parameters. The work applies the latest statistical thinking/methodology in a largely clear and careful manner. To my non-climate trained eye, the authors seem to learn things about FAMOUS that were possibly unknown before, and likely to be of interest to the community of climate modellers. In my opinion the work deserves to be published subject to a few minor changes.

I have two main criticisms of the paper. The first is that it is slightly repetitive in places. Several of the plots show very similar information, and make the same point albeit in

different ways (which may be the intention). I felt the main point of the paper could be made in less space, and that this would improve the paper.

My second criticism is that the paper is philosophically confused in places. This isn't necessarily a criticism of the paper, as most of the computer experiments community is somewhat confused about model discrepancy (as am I), but I felt the discussion lacked depth and nuance in places. Note that many of the following points are discussion rather than suggested changes to the manuscript.

**Simulator discrepancy**

As discussed, estimating simulator discrepancy is hard, as it is difficult to disentangle the effect of simulator discrepancy from the problem of estimating unknown parameters. I don't like the definition of discrepancy quoted from Williamson et al 2014, that discrepancy is an error that cannot be removed by changing the parameters without introducing more serious biases to the model. The problem is that what constitutes an acceptable discrepancy function depends upon your goal. If you aim to do prediction, then something like the above would work, as we just want to characterize the simulator error for a given parameter value. However, if the aim is to infer the parameters, and for that inference to relate to the 'true' value of those parameters, then you have to aim to model the true simulator discrepancy, which is much much harder. The problem that is hard to overcome, is that we may find the smallest simulator error occurs at parameters that are far from their 'true' values if the simulator is poor. Brynjarsdottir and O'Hagan make the point that strong prior information is needed on the true parameter values if you wish to have any hope of disentangling the parametric uncertainty from the discrepancy. I think the aim of this paper is to estimate parameters, but the approach taken is one that is perhaps better suited to prediction problems.

A discrepancy emerges in the paper, and is argued for by showing that there is an

irresolvable error. The argument used is a kind of minimum error argument: we can't simulate all four forests simultaneously, but we can do three, so let's have a discrepancy just on the Amazon, and assume the simulator is fine for the others. This sounds sensible, but it could be that the Amazon is correct and the others wrong, or that there is simulator discrepancy for all four when we use the true parameter values. I could imagine that the errors are highly correlated for the forests, so that this kind of weight of evidence approach may be flawed. This also highlights for me the weakness of this approach compared to a more traditional statistical approach. If we had statistically modelled the discrepancy, described priors, and inferred posteriors, I suspect a similar conclusion may have been reached, but the weighting would have been done using the rules of probability, and the argument would instead be over the choice of model. Here, although it is unclear to me quite how the conclusion was reached, it seems that the authors avoid the need for modelling assumptions, but instead use an informal and heuristic weighting arguments to decide where to place the discrepancy. Although they have a mechanistic explanation of why their approach makes sense, the danger is that this is done post-hoc to fit the results.

A final point on the discrepancy concerns the sentence 'We do not have enough information to create a more detailed discrepancy function: for example, one that varies across parameter space'. Why would the discrepancy vary across parameter space? I thought it was the difference between the simulator and reality when the simulator is run at the 'true' or 'best' input?

**History matching**

In the statistical part of the computer experiment community, there is an ongoing debate about whether we should do calibration or history matching (HM). I sometimes feel that HM advocates are too critical of calibration, criticising implementation problems

as if they were fundamental flaws in the framework, and conversely that the calibration crowd simply don't consider doing anything different. I like the idea of history matching, and have used it in my own work, but my understanding is that it was developed for situations where you have a huge input space, most of which is implausible, which can then mean that it is hard to accurately emulate the simulator across the entire input space. If this is the case, conservatively ruling out parts of space in a sequence of HM waves, can make emulation much easier. I have heard HM advocates then say that they might finish the analysis with a calibration, which again makes sense to me, as this can provide more nuanced information along the lines of 'we can't rule out $\theta = 2$, but it is much less likely than $\theta = 3$', which are statements that cannot be made within a HM approach. For the situation considered in this paper, there is no need to do waves of HM, as the emulator is adequate, and the data are such that only a small proportion of space can be ruled out (43% ruled out in the end). I can't help but feel that statistical calibration would have been the better approach in this case (although this is a matter of taste). Indeed, although the authors provides a brief explanation of why they prefer HM, in several places, the authors treat the output of their inference as if it were the result of a probabilistic calibration.

For example, Figure 16 is misleading. The histogram is suggestive of this being a distribution over the parameters. But as history matching was used, not calibration, there is no relevant information about the relative weighting of the parameters. This error is compounded in the sentence 'The relative frequency of NROY points is higher in some locations than others [...] suggesting a higher probability that the best estimates of the parameters is in these regions'. No statement can be made about probability here, as no probabilities were used and so this is misleading.

Line 6-8 on page 9 puzzled me, and also made me think that probabilistic calibration was perhaps what the authors had in mind. The claim is that finding the NROY region is near the edge of parameter space suggests a discrepancy function. I didn't really understand why this should be so, unless there is a secret/undeclared prior distribution

that the authors have in mind, and that they believe the parameters really lie near the middle of the a priori plausible region. Of course, in a HM approach these consideration are not taken into account.

On page 7, line 10, the authors say that the 'key' difference between calibration and HM is that points are not-ruled-out-yet (NROY) rather than 'accepted'. I find this point to be rather pedantic, as it is just a matter of labelling. I would say the key difference is that HM classifies points, but calibration describes a probability distribution over them. If we did calibration with uniform priors and thresholded the likelihood (using a pseudo-likelihood of either 0 or 1), then the two approaches can be made algorithmically equivalent (the interpretation remains different).

Finally, HM uses the implausibility given by equation 2 to score points, and then rejects points with a high score. We know from the theory of scoring rules that it is important to use a proper score, yet we can show that this score is improper (e.g. Gneiting and Raftery, JASA, 2007). Why doesn't this matter? We could use other scores in HM, and cut-offs other than the 3 sigma rule, and indeed on page 11, line 26-30, variations on how to threshold the plausibility are discussed. I support the authors' call for more research on the behaviour of the measures of implausibility, and perhaps suggest that links to scoring rules are investigated.

**Other points**

- Page 8, line 27. Where does the 0.05 observation error come from? And the sentence 'This corresponds to an expectation that the true 95% CI of $\pm$ 0.15' is incorrect I think. Pukelsheim's rule says that the 95% CI is contained within $\pm$ 0.15, not that it is equal to it. For a Gaussian rv, this would be a 99% CI for example.

- There is some confusion over the projections of points in the plots. In figure 8

for example, error is shown as a function of two parameters, where the effect of the other parameters has been averaged out. Is this useful? Just because the average error is zero, doesn't mean the error is zero anywhere. I appreciate this probably isn't what is happening, but the plots aren't necessarily a good idea.

- Page 11, line 14. I don't understand the final sentence here? According to equation 2, it makes no difference whether we assign the uncertainty to the observation or the model discrepancy. And why would we want to do this? We were told observation error was known (and fixed).

- Another point that is more discussion then criticism, as I believe it is probably common practice, is the issue of treating the climate as a static system, by spinning up the climate model to reach equilibrium. Again, I'm not a climate scientist, but as the climate is dynamic, does this practice cause a bias? Suppose we had the true simulator, with zero discrepancy, would spinning-up to equilibrium induce an error in our predictions? I appreciate there is probably no way around this.

- The language needs editing in places, with errors becoming increasingly common in later sections.

**Minor points**

- Page 1, line 10, 'find the parameters that have most impact on simulator error'. To be slightly nit-picky, I don't know what this means. Perhaps 'find the parameters that have most impact on simulator output', as simulator error, probably means the error when run at the best input.

- Page 2, line 8-11. This description is slightly confusing. Calibration, tuning, and history matching are all solving the inverse problem in some sense. Needs rephrasing, and perhaps a reference or two.

- Page 5, line 7-8. I don't believe the claim that LHC designs are better than others. I read Urban and Fricker a long time ago, but I think they just compared LHC to grid designs, and then only in empirical experiments. I'm pretty sure it is not the case that the question of the best design is settled in general (see Zhu and Stein 2006, and Zimmerman 2006 etc). I think it would be better to say that LHC designs are 'good designs'.

- Page 6, line 25, 'Gaussian' not 'gaussian'

- Page 6, line 29, 'The emulator is a nonlinear regression model' perhaps 'non-parametric' would be better than 'nonlinear', given the potential for confusion with what is normally meant by 'nonlinear regression model', i.e., non-linear in the parameters.

- Page 6, line 31. Given it is quite a long paper, there are remarkably few details about the emulator, covariance function, mean function, estimation approach etc. The review guidelines ask me to check that the paper is reproduceable, which without these details, it would not be.

- Page 8, line 31, 'We sample from the emulator uniformly across input parameter space' - this is unclear. Presumably you sampled uniformly from the input parameter space, and then from the emulator. Same again on page 11, line 2.

- Page 10, line 7, 'total effect'

- Page 12, line 29. 'that that model discrepancy uncertainty is zero'.

- Page 16, line 29/30. Rephrase sentence 'First, is there...' A dodgy emulator would lead us to think a bias exists, not cause it.
* * *

---

## Author Response (AR1)

[revised manuscript text omitted]

**Review comments and responses**

20   **Reviewer 1** (Anonymous, denoted R1:)

– R1: Review of paper: "The impact of structural error on parameter constraint in a climate model" by Doug McNeall et al. Thank you for inviting me to review this paper. The paper is interesting and important as it addresses whether a component of a GCM can be calibrated for one part of the globe, but applied elsewhere. Climate models are heavily dependent on transferability of parameterisation of sub-model structure, and a knowledge of when this fails is important.

25   I can see the aim of the paper, and it will be useful to have in the literature. However there did seem to be a slightly excessive use of statistical terminology. That's fine if the statistics is of standard form, but that's not the case here as the methods utilised are more novel. Please ensure that the literature is cited sufficiently well that any part of this paper can be understood by calling upon the appropriate referenced papers.

– Response: With a paper at the interface of climate modelling and statistics, finding the correct balance of technical versus general description will always be difficult. Our strategy was to write for a more general audience, but to include a comprehensive set of references to literature at this interface. The statistical foundations of Gaussian process emulators

are fairly standard, having been used in computer experiments across a wide range of subjects. With that in mind, we might add the following reference as a general and instructional introduction to the subject, for non-experts:

O'Hagan, A. Bayesian analysis of computer code outputs: a tutorial. Reliability Engineering & System Safety 91, no. 10 (2006): 1290-1300.

Using emulators for climate science work is rarer, although a literature is building. Our paper uses standard emulators in a less standard way, in order to learn about the model and the climate. We can expand the literature review to include more examples of emulators being used in novel ways in climate science, in order to include more context for the reader. Some specific examples of related analyses from the climate science literature are:

Williamson, D., Goldstein, M., Allison, L., Blaker, A., Challenor, P., Jackson, L. and Yamazaki, K., 2013. History matching for exploring and reducing climate model parameter space using observations and a large perturbed physics ensemble. Climate dynamics, 41(7-8), pp.1703-1729.

Bounceur, N., Crucifix, M. and Wilkinson, R.D., 2015. Global sensitivity analysis of the climate-vegetation system to astronomical forcing: an emulator-based approach. Earth System Dynamics, 6(1), p.205.

Tran, G.T., Oliver, K.I., Toal, D.J., Holden, P.B. and Edwards, N.R., 2016. Building a traceable climate model hierarchy with multi-level emulators. Advances in Statistical Climatology, Meteorology and Oceanography, 2(1), p.17.

– Action taken: We have added a section in the supplementary material with a brief outline of the emulator, more detail on the statistical modelling choices, and deeper reference to the software description paper. The introduction has been updated and streamlined, with these references added.

R1: Below are some comments that the authors might like to consider for a revised manuscript:

Overall points

– The title is possibly too general. The emphasis is on DGVM modelling of forests, not general overall issues of structure.

– Response: While we take the reviewers point here, we feel that the techniques used in the paper are sufficiently general-isable to be of interest to the wider climate modelling community. A key theme of this paper is that it attempts to improve the DVGM within the context of an Earth system model, which has it's own biases in climate simulation. An alternative title could be "The impact of structural error on parameter constraint in the land surface component of a climate model", but we welcome suggestions from the Editor.

– Action taken: None, as the Editor was happy with the original title.

– R1: The Abstract needs to be something that can be read in isolation, such that the reader can obtain a strong idea what the paper is about. To my mind, there is some repetition (e.g. three times says this uses "a history matching approach", and yet doesn't define what this actually is). Removing repetition can make space for more details. Extra description of the parameters changed would be helpful, rather than a vague "parameters that lead to a realistic forest fraction".

– Response: The reviewer makes some good points here, however describing the individual parameters in the abstract and yet making it shorter might be challenging. The focus of the paper is on the techniques for learning about the parameters, rather than the parameters themselves. Perhaps a broad description of the types of systems the parameters help control might be appropriate? We agree that the abstract could be more compact, avoid repetition and perhaps offer a clearer description of history matching. With that in mind, we suggest the following as a re-write:

We use observations of forest fraction to constrain carbon cycle and land surface input parameters of the reduced resolution global climate model, FAMOUS. We use an ensemble of climate model runs to build a computationally cheap statistical proxy (emulator) of the climate model. We then use a "history matching" approach, comparing the emulated climate model output at various parameter settings, and ruling out as implausible those where the simulated output is judged statistically incompatible with observations. We use the emulator to simulate the forest fraction at the best set of parameters implied by matching the model to the Amazon, Central African, South East Asian and North American forests in turn. We can find parameters that lead to a realistic forest fraction in the Amazon, but using the Amazon alone to tune the simulator would result in a significant overestimate of forest fraction in the other forests. Conversely, using the other forests to calibrate the model leads to a larger underestimate of the Amazon forest fraction. We argue that this finding indicates a structural model discrepancy. We characterise this discrepancy, and explore the consequences of ignoring it in a history matching exercise. We use sensitivity analysis to find the parameters which have most impact on simulator error. Finally, we perform a history matching exercise using credible estimates for simulator discrepancy and observational uncertainty terms. We are unable to constrain the parameters individually, but just under half of joint parameter space is ruled out as being incompatible with forest observations. We discuss the possible sources of the discrepancy in the simulated Amazon, including missing processes in the land surface component, and a bias in the climatology of the Amazon.

– Action taken: Abstract re-written for clarity, slightly different from this version.

– R1: Reviewing this, I'm trying to really understand what the main thrust of this paper is about, in the statistical/algorithm sense. Can I confirm that the over-arching message is that quantity delta in Eqn (1) is important, can be characterised, and shows geographical variation. To my mind, that is a powerful result. It basically says if (i) not enough process representation is introduced in to a model, then structure deficiency gets masked by parameter fitting, and (ii) doing so will create problems between different locations.

– Response: That is a good summary of the main thrust of the paper. We would also like to highlight that it is not just missing process representation, but poor process representation (i.e. biases in other parts of the climate system) that can lead to errors if the delta in equation 1 is not taken into account. We would also like to highlight some of the novel techniques that we've developed to learn about discrepancy and its impacts. We will endeavour to make this clearer in the introduction to the paper.

– Action taken: Introduction edited for clarity.

– R1: It would be nice to acknowledge that structural errors presumably also reduce confidence in any model for future projections, even when just at a single region where it performs well for contemporary periods.

– Response: I suggest that we include this point in the discussion section. One advantage of including and estimating a discrepancy term is that future projections should acknowledge the uncertainty caused by the structural discrepancy. While this may lead to more uncertain projections, they should be more robust - that is, they should offer a more accurate estimate of uncertainty.

– Action taken: This point is made in the introduction, in the simulator discrepancy section.

– R1: Page 9, starting "Does this region represent". This is a critical part of the paper, discussing how in effect a standard best-fit might not always be appropriate. Can the discussion be led back to Eqn (1), and in particular the structural delta parameter? (Also line 1, page 9, I cannot see in a Table or diagram what the alternative potential values are, for comparison against the default inputs - apologies if I've missed something). Where are the local, or continent-scale, delta values given?

– Response: At the moment, this just says "without using a structural discrepancy function", but we agree with the reviewer that this could be much clearer. We will refer this straight back to equation 1, with the implications for mean and uncertainty of the discrepancy function (not) used. The alternative potential values are a multidimensional cloud of points in parameter space, and therefore hard to summarise in a table (or even in a graphic - we are reduced to a two dimensional projection of the five dimensional space). The graphic (figure 4) has the space in normalised units - it might be clearer if we were to place the default parameters on this graphic. The model error in each forest at the input values indicated by figure 4 can be estimated by looking at figure 5, which shows the output of the model at this region.

– Action taken: Referred the section back to the equation 1. We have added the default parameter values to the graphic in figure 2, and to table 1 to make it easier for the reader to both find the values, and visualise them in comparison with the NROY regions in the other figures.

R1: Details

– P2, line 10. Again, please give the reader some idea what "History matching" is, given other quantities such as "calibration" and "tuning" are defined at this point.

– Response: We shall include an early, simple description of history matching, which may well be more unfamiliar than tuning or calibration to readers.

– Action taken: A clearer description of history matching has been added early in the introduction.

– R1: Around lines P2, lines 25-29. It would be really nice to have more concrete reasons why emulators, parameterisations etc are needed. This usually comes down to two factors: (1), computational speed prevents very high resolution

modelling, even if the processes are more fully understood. For example, parameterisation of convection. (2), we don't know what the values should be, and these may exhibit strong regional heterogeneity. The latter is more the case for this paper, with questions asked as to what are the appropriate number of plant functional types that should be in land surface models - and if the number is high, can for example EO provide the values.

– Response: The reviewer makes a good point that we don't discuss the possibility of regionally varying parameters, and what that means for the current analysis. We shall include a section on regionally varying parameters in the discussion section, and expand the section on paramaterisation accordingly.

– Action taken: The suggested examples have been added to the introduction section.

– R1: Check notation is consistent throughout. P3, line 23, FAMOUS is described as a "climate simulator". In the minds of the authors, is this different to a standard GCMs. Do they regard FAMOUS's reduced resolution as removing it from being regarded as a full GCM?

– Response: In the statistical emulator literature "simulator" is often used for computational process models in order to distinguish them from statistical models. We will make this clear, and review use of "simulator" and "model" in the paper in order to ensure consistency.

– Action taken: Added footnote to this effect, and use of "simulator" is more standardised through the text now.

– R1: Again, in Section 1.3, this is now the 7th or 8th time that "history matching" is mentioned - it would be good to help the reader as to what it is, even if it is only to provide a methodological citation at this point.

– Response: See previous response.

– Action taken: Earlier history matching description added.

– R1: Cox (2001) is a technical note. Better to give a peer-reviewed reference?

– Response: It is possible to cite well known papers that use TRIFFID (e.g. Cox 2000), but Cox (2001) is the the standard reference that outlines the technical details of TRIFFID.

Cox, P.M., Betts, R.A., Jones, C.D., Spall, S.A. and Totterdell, I.J., 2000. Acceleration of global warming due to carbon-cycle feedbacks in a coupled climate model. Nature, 408(6809), pp.184-187.

– Action taken: none.

– R1: P5, line 1. I don't understand the context of the sentence: "The Amazon region is not wet enough for a fully humid region to exist". If this refers to the FAMOUS model, and in particular its atmospheric response, then this will make any DGVM fail if rainfall totals are too small. P5, discussion of beta parameter. In a similar vain to the comment above, is it OK to treat the atmospheric beta parameter as a "nuisance" parameter? Isn't there a risk that errors in GCM-projected precipitation - for example - will affect best-fit parameters in Table 1?

- Response: FAMOUS has known biases, including a climatologically dry Amazon region, and this is indeed one of the strong candidates for low forest fractions in that region, as discussed later in the paper. However, in the Amazon region there are also possible confounding feedbacks between land cover and climate, making attribution of any biases more difficult. No climate simulation is perfect, and biases large or small are a common problem to be dealt with in any analysis. Our analysis offers new techniques to identify and characterise such biases, and the way that they might impact our estimates of the values of input parameters. The beta parameter is not correlated with any of the land surface parameters in the ensemble design, and so we felt justified in excluding it from analysis of the land surface parameters. However, it may well have an impact on climatology, and this could be the subject of a future study.

- Action taken: None.

- R1: P5, line 18. From code that is shared with other centres, TRIFFID has a rapid spin-up option to near-equilibrium. Does it really need 10000 years?

- Response: The fast spin-up mode was used in the simulations - only the equivalent of 10,000 years for each decade was used in this mode. The climate simulations were the averages of the last 30 years of a 200 year run. We shall make this clearer in the text.

- Action taken: text amended to reflect the reviewer's suggestion.

- R1: Trivial thing, but it might be nice in Figure 2 to write as S.E.Asia (not SEASIA).

- Response: This will be amended to be consistent with the other plots (a space added).

- Action taken: None, as the headings are directly taken from the R data frame that contains the data. Keeping a "no spaces" name ensures consistency with all of the other parameters in this diagram, and offers a direct check that we are plotting the correct thing.

- R1: Can I confirm that a reader could find all details of the emulator in the Roustant et al 2012 paper. So, for instance, what a "leave-one-out cross validation metric" is.

- Response: Roustant et al. (2012) is very comprehensive in its mathematical description of the emulator, and the software package that it informs. Leave-one-out cross validation is not related to the emulator itself, but is a broader validation algorithm. We will include a suitable reference (e.g. Hastie, Tibshirani and Friedman (2001).

  Hastie, T., Tibshirani, R. and Friedman, J. 2001. The elements of statistical learning (Vol. 1). Springer, Berlin: Springer series in statistics.

- Action taken: [as above] We have added a section in the supplementary material with a brief outline of the emulator, more detail on the statistical modelling choices, and deeper reference to the software description paper. We have added the latest version of the reference (Hastie et al 2009) to cross validation, and other statistical model verification techniques.

- R1: Figure 7 I find very useful as it allows assessment of the geographical differences, providing more information that the global parameterisation Table 3. There are quite a few statistical methods available to determine parameter importance and/or nuisance parameters. An extra sentence stating what additional benefit the FAST algorithm brings would be helpful - i.e. beyond just the Saltelli reference.

- Response: The FAST algorithm is ideally suited to our situation in that a) it provides an accurate global sensitivity analysis, including main effects and interaction terms and b) is easily and cheaply calculated using the emulator and a convenient R package. We shall include a sentence to this effect in the section.

- Action taken: Further justification for using the FAST algorithm added.

- R1: Figure 8 is important as it shows how the Amazon has a difference response. Or put another way, a calibration of NL0 and V_CRIT_ALPHA for the Amazon could find a pair of parameters that would clearly be sub-optimal when applied to the other 3 regions. And vice-versa. I'd like to see more discussion around Figure 8, how it demonstrates the structural problems (i.e. very different responses to NL0 and V_CRIT_ALPHA, depending on location), and again - can this be related back to the delta parameter? This will also link better to the paper title, which is about model structural problems.

- Response: Linking this clearly back to the structural discrepancy function at this point is a good idea. However, the discussion that the reviewer requests here is a large part of the later analysis (e.g. figures 10 - 12, and section 3.5). We could indicate the more detailed discussion in this later section in the text of this earlier section.

- Action taken: We've clarified the fact that non-overlapping regions of zero error in this reduced input space makes it likely we'll need a discrepancy function.

- R1: Figure 13 is nice and clear, and in many ways it is a shame that the paper is so long in technical details before getting to that point. Obviously this is a slightly naive comment, but could it simply be that the trees of the Amazon have evolved differently to those of Africa. This could possibly be due to different imposed climatologies that the trees have adapted/acclimated to. So one conclusion of this paper could simply be that any land surface model such as TRIFFID requires a parameter mask, or ancillary fields, that are different for different places. The paper hints at this, page 16, in "Causes of discrepancy", where different rooting depths are considered. One future work extension might therefore be to include a root depth as a geographically-varying parameter, to add to those in Table 1? Would this then collapse delta down to zero for all locations?

- Response: This interesting and useful idea should clearly be included in the discussion section.

- Action taken: This suggestion is included in the discussion, under the "Causes of discrepancy" section.

**Reviewer 2 (Richard Wilkinson, RW)**

– RW: This paper describes a thorough and detailed investigation into the ability of FAMOUS to predict forest fraction. The paper starts from the pretext of being given an ensemble of pre-run simulator evaluations and observation data corresponding to some of the outputs, and being asked to estimate some of the parameters. The work applies the latest statistical thinking/methodology in a largely clear and careful manner. To my non-climate trained eye, the authors seem to learn things about FAMOUS that were possibly unknown before, and likely to be of interest to the community of climate modellers. In my opinion the work deserves to be published subject to a few minor changes.

I have two main criticisms of the paper. The first is that it is slightly repetitive in places. Several of the plots show very similar information, and make the same point albeit in different ways (which may be the intention). I felt the main point of the paper could be made in less space, and that this would improve the paper. My second criticism is that the paper is philosophically confused in places. This isn't necessarily a criticism of the paper, as most of the computer experiments community is somewhat confused about model discrepancy (as am I), but I felt the discussion lacked depth and nuance in places. Note that many of the following points are discussion rather than suggested changes to the manuscript.

– Response: Richard makes some valid points here, but the paper is long because it shows a number of novel analysis techniques, each of which provide some unique information about the simulator, its errors, and the relationship between the input parameters and the simulator output. Finding a clear narrative that included these analyses was a challenge, but valuable. Excluding some of these analyses may well make the paper clearer in its main message, but at the risk of changing the focus on explanatory analyses, which I feel is a strength of the paper. However, I think it would be possible to move some of the analyses to supplementary material, if that was deemed beneficial. Sections such as 3.2 (sensitivity analysis) and 3.4 (How much space is ruled out by combinations of observations?), and parts of section 3.5 (e.g. figure 11), are somewhat additional to the main arguments of the paper, and could be moved. Both reviewers have made suggestions for expanding the discussion, which it is hoped will add depth and nuance.

– Actions taken: Removed discussion of sum of sensitivity effects (and related table 4), as it is a distraction. R1 says that more specific sensitivity is useful, so that section remains. Each section has been reviewed, to tighten language and to purge repetition. Some plots and analysis have been removed, and some moved to supplementary material, so the main arguments of the paper are now delivered in a shorter paper.

– RW: Simulator discrepancy. As discussed, estimating simulator discrepancy is hard, as it is difficult to disentangle the effect of simulator discrepancy from the problem of estimating unknown parameters. I dont like the definition of discrepancy quoted from Williamson et al 2014, that discrepancy is an error that cannot be removed by changing the parameters without introducing more serious biases to the model. The problem is that what constitutes an acceptable discrepancy function depends upon your goal. If you aim to do prediction, then something like the above would work, as we just want to characterize the simulator error for a given parameter value. However, if the aim is to infer the parameters, and for that inference to relate to the "true" value of those parameters, then you have to aim to model the true simulator discrepancy, which is much much harder. The problem that is hard to overcome, is that we may find the smallest simulator error occurs at parameters that are far from their "true' values if the simulator is poor. Brynjarsdottir and O'Hagan make the

point that strong prior information is needed on the true parameter values if you wish to have any hope of disentangling the parametric uncertainty from the discrepancy. I think the aim of this paper is to estimate parameters, but the approach taken is one that is perhaps better suited to prediction problems.

A discrepancy emerges in the paper, and is argued for by showing that there is an irresolvable error. The argument used is a kind of minimum error argument: we can't simulate all four forests simultaneously, but we can do three, so let's have a discrepancy just on the Amazon, and assume the simulator is fine for the others. This sounds sensible, but it could be that the Amazon is correct and the others wrong, or that there is simulator discrepancy for all four when we use the true parameter values. I could imagine that the errors are highly correlated for the forests, so that this kind of weight of evidence approach may be flawed. This also highlights for me the weakness of this approach compared to a more traditional statistical approach. If we had statistically modelled the discrepancy, described priors, and inferred posteriors, I suspect a similar conclusion may have been reached, but the weighting would have been done using the rules of probability, and the argument would instead be over the choice of model. Here, although it is unclear to me quite how the conclusion was reached, it seems that the authors avoid the need for modelling assumptions, but instead use an informal and heuristic weighting arguments to decide where to place the discrepancy. Although they have a mechanistic explanation of why their approach makes sense, the danger is that this is done post-hoc to fit the results.

A final point on the discrepancy concerns the sentence "We do not have enough information to create a more detailed discrepancy function: for example, one that varies across parameter space". Why would the discrepancy vary across parameter space? I thought it was the difference between the simulator and reality when the simulator is run at the "true" or "best" input?

– Response: The aim of this analysis is indeed to find good parameter sets, but also to use information that comes from the analysis to characterise the simulator discrepancy, and its consequences. It should perhaps be seen as a valuable, and useful step along the road towards a full statistical calibration treatment of the problem, rather than an end point. History matching is conceptually simple, easy to code, and fast to calculate, making it an accessible and attractive option for introducing more statistically robust analyses. With this in mind, we could expand the discussion to include some of the points made by Richard here - particularly the advantages of using a full calibration, compared to our more ad hoc approach.

There is also no doubt that, even without a full calibration exercise, there is information that can be used to make judgements within the history matching framework. If we know that the model contains climate biases, and that parameters are the result of a long modelling effort and knowledge, do we rule them out as implausible when the model does not reproduce the Amazon? Very likely we would not, especially when other forests are adequately modelled.

Regarding the discrepancy varying across parameter space - this is incorrect, and we should remove the statement.

– Actions taken: Statement about variation across parameter space removed from introduction. Added an alternative definition of discrepancy under the best input approach, from e.g. Goldstein et al (2009) to introduction. Added the point

that strong priors are needed to distinguish parameter uncertainty from discrepancy. Included a section in the discussion suggesting that a full probabilistic calibration could have advantages over our approach, particularly with regard to weighting inputs.

– RW: History matching. In the statistical part of the computer experiment community, there is an ongoing debate about whether we should do calibration or history matching (HM). I sometimes feel that HM advocates are too critical of calibration, criticising implementation problems as if they were fundamental flaws in the framework, and conversely that the calibration crowd simply don't consider doing anything different. I like the idea of history matching, and have used it in my own work, but my understanding is that it was developed for situations where you have a huge input space, most of which is implausible, which can then mean that it is hard to accurately emulate the simulator across the entire input space. If this is the case, conservatively ruling out parts of space in a sequence of HM waves, can make emulation much easier. I have heard HM advocates then say that they might finish the analysis with a calibration, which again makes sense to me, as this can provide more nuanced information along the lines of "we can't rule out $\theta = 2$, but it is much less likely than $\theta = 3$", which are statements that cannot be made within a HM approach. For the situation considered in this paper, there is no need to do waves of HM, as the emulator is adequate, and the data are such that only a small proportion of space can be ruled out (43% ruled out in the end). I can't help but feel that statistical calibration would have been the better approach in this case (although this is a matter of taste). Indeed, although the authors provides a brief explanation of why they prefer HM, in several places, the authors treat the output of their inference as if it were the result of a probabilistic calibration.

– Response: Again, we feel that history matching has something to offer as an accessible alternative to, and preparation for, full calibration. It is also a good platform for some of the "what if" type exploratory analyses that we conduct in the study.

– Action taken: See below.

– RW: For example, Figure 16 is misleading. The histogram is suggestive of this being a distribution over the parameters. But as history matching was used, not calibration, there is no relevant information about the relative weighting of the parameters. This error is compounded in the sentence "The relative frequency of NROY points is higher in some locations than others [...] suggesting a higher probability that the best estimates of the parameters is in these regions". No statement can be made about probability here, as no probabilities were used and so this is misleading.

– Response: In hindsight, Richard is right here, and we should make more effort to make sure that our analyses are not interpreted as fully probabilistic. This will include removing figure 16 and associated text, and clarifying figure captions containing histograms.

– Action taken: Clarified the description of history matching, and its uses, to distinguish it better from calibration. Removed Figure 16 and associated text. Added a section in the discussion, pointing out that calibration would offer tools to move beyond the challenges described in this paper.

– RW: Line 6-8 on page 9 puzzled me, and also made me think that probabilistic calibration was perhaps what the authors had in mind. The claim is that finding the NROY region is near the edge of parameter space suggests a discrepancy function. I didn't really understand why this should be so, unless there is a secret/undeclared prior distribution that the authors have in mind, and that they believe the parameters really lie near the middle of the a priori plausible region. Of course, in a HM approach these consideration are not taken into account.

– Response: This argument may be the result of two things: 1) the structure of the study, where a pre-computed ensemble has been passed along to a (mostly) new set of authors, with little or no opportunity to re-run. In this case, the ensemble range is being used as the de facto plausible range of parameter values, which is perhaps not what was intended. This might mean that plausible parameter settings are to be found outside of the initial parameter space, and that plausible parameters are found against the very edge of parameter space. 2) As in this case, there is a suspicion that there is a discrepancy (a low Amazon forest fraction, perhaps caused by a climate bias), and some default parameters near the centre of the space, but no firm evidence until the analysis is run. The modellers then have to make a judgement as to whether applying a discrepancy term, or excluding the default parameters is appropriate, which is explored in the discussion section.

– Action take: Problematic sentences removed.

– RW: On page 7, line 10, the authors say that the "key" difference between calibration and HM is that points are not-ruled-out-yet (NROY) rather than "accepted". I find this point to be rather pedantic, as it is just a matter of labelling. I would say the key difference is that HM classifies points, but calibration describes a probability distribution over them. If we did calibration with uniform priors and thresholded the likelihood (using a pseudo- likelihood of either 0 or 1), then the two approaches can be made algorithmically equivalent (the interpretation remains different).

– Response: This point is well made, and we shall amend the text.

– Action taken: Text amended to clarify differences between history matching and calibration.

– RW: Finally, HM uses the implausibility given by equation 2 to score points, and then rejects points with a high score. We know from the theory of scoring rules that it is important to use a proper score, yet we can show that this score is improper (e.g. Gneiting and Raftery, JASA, 2007). Why doesn't this matter? We could use other scores in HM, and cut-offs other than the 3 sigma rule, and indeed on page 11, line 26-30, variations on how to threshold the plausibility are discussed. I support the authors' call for more research on the behaviour of the measures of implausibility, and perhaps suggest that links to scoring rules are investigated.

– Action taken - none

Other points

– RW: Page 8, line 27. Where does the 0.05 observation error come from? And the sentence "This corresponds to an expectation that the true 95% CI of ±0.15" is incorrect I think. Pukelsheim's rule says that the 95% CI is contained within ±0.15, not that it is equal to it. For a Gaussian rv, this would be a 99% CI for example.

– Response: The 0.05 observation error is an expert judgement of the true observational uncertainty, and a useful illustrative value, given that there is little information on the uncertainty of the observations themselves. We will make this clear in the text.

– Action taken: Now reads "On the advice of domain experts, we assume observational uncertainty of 0.05 (one standard deviation) in the Amazon, Central African, South East Asian and North American forests as broadly representative, or at least usefully illustrative. This corresponds to an expectation that the true 95% confidence interval is contained within the interval of ±0.15, following Pukelsheim's rule. This is nearly a third of the available range of zero to one, and we contend that it would be hard to argue that this represents an over-constraint."

– RW: There is some confusion over the projections of points in the plots. In figure 8 for example, error is shown as a function of two parameters, where the effect of the other parameters has been averaged out. Is this useful? Just because the average error is zero, doesn't mean the error is zero anywhere. I appreciate this probably isn't what is happening, but the plots aren't necessarily a good idea.

– Response: The scatter (in colour) of the plots gives a visual impression of how much the error varies across the other parameters, and we would argue that the plots give a good indication of the regions of likely small error, even if they do not show where (or if) the error is exactly zero.

– Action taken: None

– RW: Page 11, line 14. I don't understand the final sentence here? According to equation 2, it makes no difference whether we assign the uncertainty to the observation or the model discrepancy. And why would we want to do this? We were told observation error was known (and fixed).

– Response: We should make it clearer that the observational error is assumed, an expert judgement, and that arguments could be made for other values.

– Action taken: text amended to clarify that the observational and structural discrepancy uncertainty are assumed in this part of the experiment, but that the emulator uncertainty is emergent.

– RW: Another point that is more discussion then criticism, as I believe it is probably common practice, is the issue of treating the climate as a static system, by spinning up the climate model to reach equilibrium. Again, I'm not a climate scientist, but as the climate is dynamic, does this practice cause a bias? Suppose we had the true simulator, with zero discrepancy, would spinning-up to equilibrium induce an error in our predictions? I appreciate there is probably no way around this.

- Response: The practice of spinning up a model is a useful way to remove biases, given that we very likely do not have adequate data to initialise the entire state of the climate system. A spin up to some historical state, followed by a period of historical forcing can be used to get the state and the dynamics of the system correct, before predictions are made.

- Action taken: none.

- RW: The language needs editing in places, with errors becoming increasingly common in later sections.

- Action taken: Language was reviewed, corrected and simplified throughout the manuscript.

Minor points

- RW: Page 1, line 10, "find the parameters that have most impact on simulator error". To be slightly nit-picky, I don't know what this means. Perhaps "find the parameters that have most impact on simulator output", as simulator error, probably means the error when run at the best input.

- Response: This was an attempt to be compact, but we will correct to make this clearer.

- Action taken: language clarified - now suggest that we find those parameters that have the largest impact on simulator output.

- RW: Page 2, line 8-11. This description is slightly confusing. Calibration, tuning, and history matching are all solving the inverse problem in some sense. Needs rephrasing, and perhaps a reference or two.

- Response: We will rephrase this section as suggested.

- Action taken: The section was rephrased for clarity.

- RW: Page 5, line 7-8. I don't believe the claim that LHC designs are better than others. I read Urban and Fricker a long time ago, but I think they just compared LHC to grid designs, and then only in empirical experiments. I'm pretty sure it is not the case that the question of the best design is settled in general (see Zhu and Stein 2006, and Zimmerman 2006 etc). I think it would be better to say that LHC designs are "good designs".

- Response: We will rephrase this section as suggested.

- Action taken: We have stated simply that these designs are commonly used to construct emulators.

- RW Page 6, line 25, "Gaussian' not "gaussian"

- RW Page 6, line 29, "The emulator is a nonlinear regression model' perhaps "non-parametric" would be better than "nonlinear", given the potential for confusion with what is normally meant by "nonlinear regression model" i.e., non-linear in the Parameters.

- Response: We will rephrase as suggested.

- Action taken: both rephrased.

- RW Page 6, line 31. Given it is quite a long paper, there are remarkably few details about the emulator, covariance function, mean function, estimation approach etc. The review guidelines ask me to check that the paper is reproduceable, which without these details, it would not be.

- Response: These details could go in the supplementary material, along with the emulator verification.

- Action taken: These details are now in the supplementary material.

- RW: Page 8, line 31, "We sample from the emulator uniformly across input parameter space" - this is unclear. Presumably you sampled uniformly from the input parameter space, and then from the emulator. Same again on page 11, line 2.

- Action taken: rephrased for clarity

- RW: Page 10, line 7, "total effect"

- RW: Page 12, line 29. "that that model discrepancy uncertainty is zero".

- RW: Page 16, line 29/30. Rephrase sentence "First, is there..." A dodgy emulator would lead us to think a bias exists, not cause it.

- Response: We will rephrase as suggested.

- Action taken: All rephrased as suggested.

*Author contributions.* DM and all authors designed the analysis. DM conducted the analysis and wrote the paper. JW provided the FAMOUS ensemble and BB provided the observed forest fraction data.

*Acknowledgements.* This work was supported by the Joint UK BEIS/Defra Met Office Hadley Centre Climate Programme (GA01101). DM was supported on secondment to Exeter University by the Met Office Academic Partnership (MOAP) for part of the work. JW was
20 supported by funding from Statoil ASA, Norway. RB is a member of the editorial board of Earth System Dynamics.

The works published in this journal are distributed under the Creative Commons Attribution 3.0 License. This licence does not affect the Crown copyright work, which is re-usable under the Open Government Licence (OGL). The Creative Commons Attribution 3.0 License and the OGL are interoperable and do not conflict with, reduce or limit each other. ©Crown copyright 2016

[revised manuscript text omitted]